# A structural mechanism for directing corepressor-selective inverse agonism of PPARγ

Richard Brust [1], Jinsai Shang [1], Jakob Fuhrmann[1], Sarah A. Mosure[1,2], Jared Bass[1], Andrew Cano[1,3], Zahra Heidari[4,5], Ian M. Chrisman[5,6], Michelle D. Nemetchek[5,6], Anne-Laure Blayo[7], Patrick R. Griffin[1,7], Theodore M. Kamenecka[7], Travis S. Hughes[4,5,6] & Douglas J. Kojetin [1,7]

Small chemical modifications can have significant effects on ligand efficacy and receptor activity, but the underlying structural mechanisms can be difficult to predict from static crystal structures alone. Here we show how a simple phenyl-to-pyridyl substitution between two common covalent orthosteric ligands targeting peroxisome proliferator-activated receptor (PPAR) gamma converts a transcriptionally neutral antagonist (GW9662) into a repressive inverse agonist (T0070907) relative to basal cellular activity. X-ray crystallography, molecular dynamics simulations, and mutagenesis coupled to activity assays reveal a water-mediated hydrogen bond network linking the T0070907 pyridyl group to Arg288 that is essential for corepressor-selective inverse agonism. NMR spectroscopy reveals that PPARγ exchanges between two long-lived conformations when bound to T0070907 but not GW9662, including a conformation that prepopulates a corepressor-bound state, priming PPARγ for high affinity corepressor binding. Our findings demonstrate that ligand engagement of Arg288 may provide routes for developing corepressor-selective repressive PPARγ ligands.

[1] Department of Integrative Structural and Computational Biology, The Scripps Research Institute, Jupiter, FL 33458, USA. [2] Skaggs Graduate School of Chemical and Biological Sciences, The Scripps Research Institute, Jupiter, FL 33458, USA. [3] High School Student Summer Internship Program, The Scripps Research Institute, Jupiter, FL 33458, USA. [4] Department of Biomedical and Pharmaceutical Sciences, University of Montana, Missoula, MO 59812, USA. [5] Center for Biomolecular Structure and Dynamics, University of Montana, Missoula, MT 59812, USA. [6] Biochemistry and Biophysics Graduate Program, University of Montana, Missoula, MT 59812, USA. [7] Department of Molecular Medicine, The Scripps Research Institute, Jupiter, FL 33458, USA. Correspondence and requests for materials should be addressed to D.J.K. (email: dkojetin@scripps.edu)

The nuclear receptor peroxisome proliferator-activated receptor gamma (PPARγ) is a target for antidiabetic drugs, including the thiazolidinedione (TZD) class of molecules[1]. TZDs are full agonists of PPARγ that promote transcription of PPARγ target genes. Unfortunately, therapeutic use of TZDs has adverse side effects, including brittle bones from the differentiation of bone into fat. Although originally it was thought that full activation of PPARγ was required for anti-diabetic efficacy, recent studies have shown that antidiabetic PPARγ ligands alter posttranslational modifications impacting target gene expression. These ligands have been shown to span a wide range of efficacies—including full and partial agonists, antagonists, and inverse agonists that have robust or mild acti-vating, neutral, or repressive transcriptional properties, respec-tively[2–5]. Importantly, repressive PPARγ modulators promote bone formation[5,6], and pharmacological repression or antagon-ism of PPARγ is implicated in the treatment of obesity[7,8] and cancer[9–12].

In order to understand how to harness the positive effects of targeting PPARγ, we need to better understand the structural mechanisms that elicit PPARγ activation (agonism) and repres-sion (inverse agonism) relative to basal cellular activity. The distinct pharmacological phenotypes of PPARγ ligands are dic-tated by ligand-dependent recruitment of transcriptional cor-egulator proteins (coactivators and corepressors) to the PPARγ ligand-binding domain (LBD). The LBD contains the orthosteric ligand-binding pocket, which is the binding site for endogenous and most synthetic ligands, as well as the activation function-2 (AF-2) coregulator binding surface. The AF-2 surface is com-posed of three LBD structural elements: helix 3, helix 5, and the critical helix 12 that moves between two or more conformations in the absence of ligand[13–15].

The structural mechanisms affording PPARγ activation are well understood. Agonists stabilize an active state of the AF-2 surface by forming hydrogen bonds with residues near helix 12. Full agonists form a critical hydrogen bond with the side chain of Y473 on helix 12 to strengthen coactivator/weaken corepressor binding affinities, inducing transcriptional activation[13,15,16]. Partial agonists generally do not hydrogen bond to Y473, but mildly stabilize helix 12 via interactions with other regions of the ligand-binding pocket, resulting in less pronounced changes in coregulator affinity and transcriptional activation[15–17]. Partial agonism can also be elicited by ligands that hydrogen bond to Y473 but make unfavorable contacts to nearby residues such as Q286 on helix 3 or other nearby regions[18,19]. Antagonists, which make unfavorable interactions with F282 on helix 3, do not sta-bilize helix 12 and display negligible changes in activation[3]. These findings have established the structural mechanisms for eliciting robust (agonist), weak (partial agonist), or no (antagonist) tran-scriptional activation of PPARγ. An inverse agonist could display a profile opposite of an agonist, increasing the binding affinity of corepressors and decreasing the binding affinity of coactivators, or weaken the affinity for coactivators or coregulators, resulting in transcriptional repression. However, relatively few studies have explored the structural mechanisms by which ligands repress PPARγ transcription[6,20], and it remains poorly understood how to design inverse agonists.

Here we compare two commonly used covalent PPARγ ligands, GW9662[21] and T0070907[22], which were originally defined as antagonists not because of their effects on PPARγ transcription, but because they covalently attach to C285 within the orthosteric ligand-binding pocket and physically block other ligands from binding PPARγ. Remarkably, despite differing by only a simple methine (CH) to nitrogen substitution, T0070907 represses PPARγ transcription compared to GW9662, which shows negligible effects on basal transcription[21–23]. Crystal structures of PPARγ bound to T0070907 or GW9662 reveal no major overall structural differences that explain the difference in efficacy. However, a water-mediated hydrogen bond network that uniquely links R288 to the T0070907 pyridyl group—an inter-action that cannot occur with GW9662, which lacks a hydrogen bond acceptor—is essential for corepressor-selective cellular repression of PPARγ. NMR analysis shows that T0070907-bound PPARγ populates two long-lived structural conformations, one of which resembles the state populated by GW9662 and a unique state that is similar to the corepressor-bound state, thus revealing a structural mechanism for directing corepressor-selective repression of PPARγ.

## Results

**T0070907 lowers basal activity of PPARγ.** GW9662 and T0070907 (Fig. 1a) contain the same 2-chloro-5-nitro-*N*-phe-nylbenzamide scaffold but differ by a simple atom change: a ring carbon in the GW9662 phenyl group is replaced by a nitrogen-containing pyridyl group in T0070907. The original discovery of GW9662 and T0070907 defined these ligands as covalent antagonists because they covalently bind to C285 in the orthos-teric pocket and inhibit rosiglitazone, a noncovalent agonist, from binding and activating PPARγ[21,22]. However, these antagonist definitions were not pharmacological descriptions of their specific activities on PPARγ (Supplementary Table 1). We compared GW9662 and T0070907 to other noncovalent activating and repressive PPARγ compounds (Supplementary Figure 1) using a cell-based transcription assay sensitive to activating and repres-sive compounds (Fig. 1b) due to the increased basal activity that occurs upon transfection of full-length PPARγ in the absence of exogenous ligand (Supplementary Figure 2). Opposite to the agonist rosiglitazone that increased PPARγ transcription and consistent with a Gal4-based assay in its original report[22], T0070907 repressed PPARγ transcription relative to vehicle-treated cells, while GW9662 did not significantly affect PPARγ transcription. The repressive efficacy of T0070907 is similar to or better than the non-covalent inverse agonists SR10221 and SR2595, respectively, which are analogs of the non-covalent antagonist parent compound SR1664[6]. The activating and repressive trends were observed in cells cultured in a media containing normal fetal bovine serum (FBS) as well as charcoal stripped FBS (Supplementary Figure 2), indicating the exogenous lipids and fatty acids present in the cell culture media do not have a significant influence on the pharmacological activities of the activating and repressive synthetic ligands tested. This observa-tion is consistent with work showing that a PPARγ mutant unable to be activated by synthetic or endogenous ligands but still dis-played significant transcriptional activity similar to wild-type receptor[24]. Furthermore, the ligands did not significantly affect PPARγ protein levels (Supplementary Figure 3) and were not cytotoxic (Supplementary Figure 4) at the concentrations tested (5 μM), indicating the ligand pharmacological activities, particu-larly for the repressive ligands, are not due to reduced protein levels or cell death. We confirmed the ligand activity profiles using qPCR analysis of 3T3-L1 preadipocyte cells treated with the ligands (Fig. 1c), which revealed similar activating, neutral, and repressive efficacy trends on the expression levels of PPARγ target genes involved in adipocyte differentiation (*FABP4* and *CD36*).

We next characterized how the ligands affect the recruitment of peptides derived from the TRAP220 coactivator and the NCoR1 corepressor (Fig. 2a), two coregulator proteins that influence PPARγ-mediated transcription[25,26], to the PPARγ LBD. Compared to unliganded apo-PPARγ LBD (Fig. 2b), nonanoic acid (an endogenous PPARγ agonist)[27] and to a larger degree rosiglitazone (a synthetic PPARγ agonist) increased the affinity of

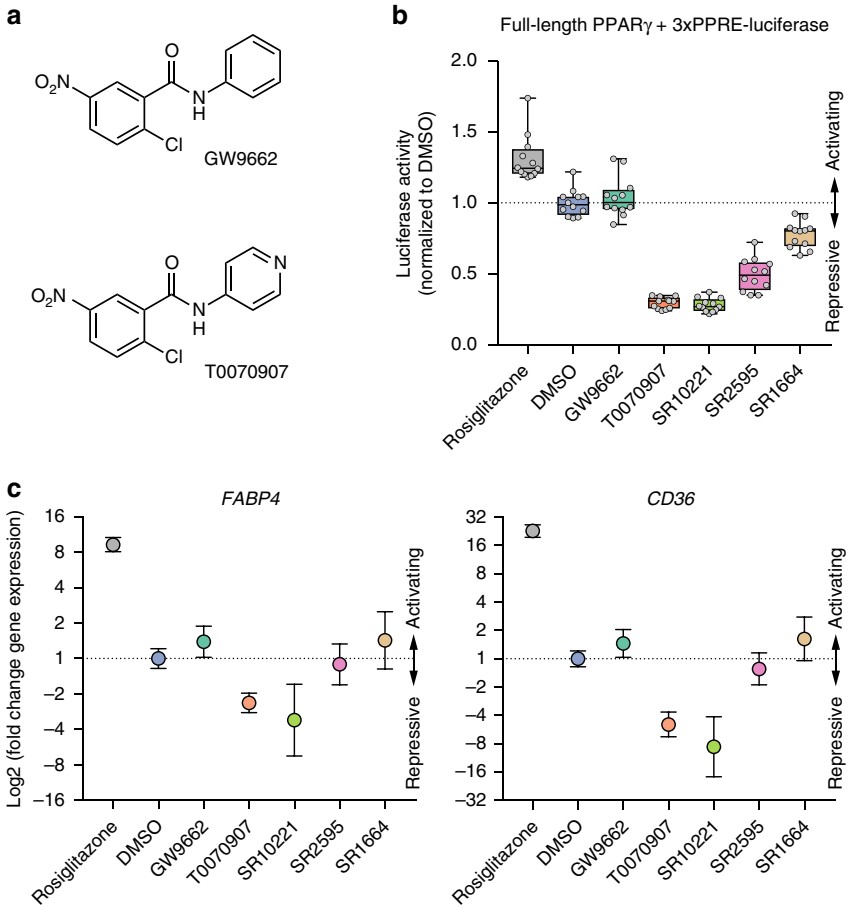

**Fig. 1** Cellular profiling of activating, neutral, and repressive synthetic PPARγ ligands. **a** Chemical structures of GW9662 and T0070907. **b** Cell-based full-length PPARγ luciferase transcriptional assay showing the effect of activating, neutral, and repressive PPARγ ligands (5 μM) on full-length PPARγ transcription in HEK293T cells. Individual points ($n = 12$) normalized to DMSO control (mean) are plotted as white circles on top of a box-and-whiskers plot; the box represents 25th, median, and 75th percentile of the data, and the whiskers plot the entire range of values. **c** Relative quantitation of fold change in PPARγ target gene expression determined by qPCR normalized to TBP expression of 3T3-L1 cells treated with transcriptionally activating, neutral, or repressive PPARγ ligands (10 μM), plotted as relative log2 of data calculated using the $2^{-\Delta\Delta Ct}$ method ($n = 3$) with error bars representative of the upper and lower limits. Data are representative of at least 3 independent experiments

TRAP220 and decreased the affinity of NCoR1. T0070907 displayed an opposite coregulator affinity profile compared to the agonists; we observed the same NCoR1 affinity trends for full-length PPARγ (Supplementary Figure 5). GW9662 caused increased affinity for NCoR1, but unlike T0070907 it did not weaken TRAP220 affinity, which may contribute to its transcriptionally neutral cellular profile. Interestingly, the repressive compounds SR2595 and SR10221 decreased affinity for both TRAP220 and NCoR1. The original report of T0070907 showed that it increased binding of a peptide derived from the NCoR1 corepressor to the PPARγ LBD[22] and increased binding of full-length NCoR1 to full-length PPARγ/RXRα heterodimer bound to DNA. Using a time-resolved fluorescence resonance energy transfer (TR-FRET) assay, we similarly found that T0070907 increases the interaction of the NCoR1 peptide to the PPARγ LBD and to full-length PPARγ/RXRα heterodimer alone or bound to DNA (Supplementary Figure 6), characteristic of an inverse agonist.

Given that the potency of direct covalent binding of GW9662 and T0070907 to PPARγ are similar (Supplementary Figure 7)[12], we were interested in the structural mechanism by which the seemingly minor methine-to-nitrogen ligand substitution could switch a covalent antagonist (GW9662) into a covalent inverse agonist (T0070907). Noncovalent repressive (inverse agonist)

analogs of the PPARγ agonist farglitazar[20] and the antagonist SR1664[6] were designed to perturb the conformation of the helix 12/AF-2 surface. The farglitazar analogs contain ligand extensions towards helix 12. SR2595 and SR10221 contain a tert-butyl extension that perturbed the conformation of F282 on helix 3 (Fig. 2c) located within the orthosteric pocket near the loop preceding helix 12, a region we refer to as the helix 12 subpocket. We previously showed that the F282/AF-2 steric clash caused by SR2595 and SR10221 binding increases the dynamics of helix 3 and 12, which are part of the AF-2 surface, thereby reducing the basal activity of PPARγ[6]. Our coregulator recruitment data show this AF-2 clash weakens coactivator and corepressor affinity, causing transcriptional repression. However, the phenyl (GW9662) to pyridyl (T0070907) change is distant from F282, the helix 12 subpocket, or the AF-2 surface (Fig. 2c), suggesting the corepressor-selective inverse agonist transcriptionally repressive profile of T0070907 may originate from a previously uncharacterized structural mechanism.

**A pyridyl-water hydrogen bond network unique to T0070907.** To gain insight into T0070907's structural mechanism of action, we solved the crystal structure of the PPARγ LBD covalently bound to T0070907 to a resolution of 2.26 Å (PDB code 6C1I;

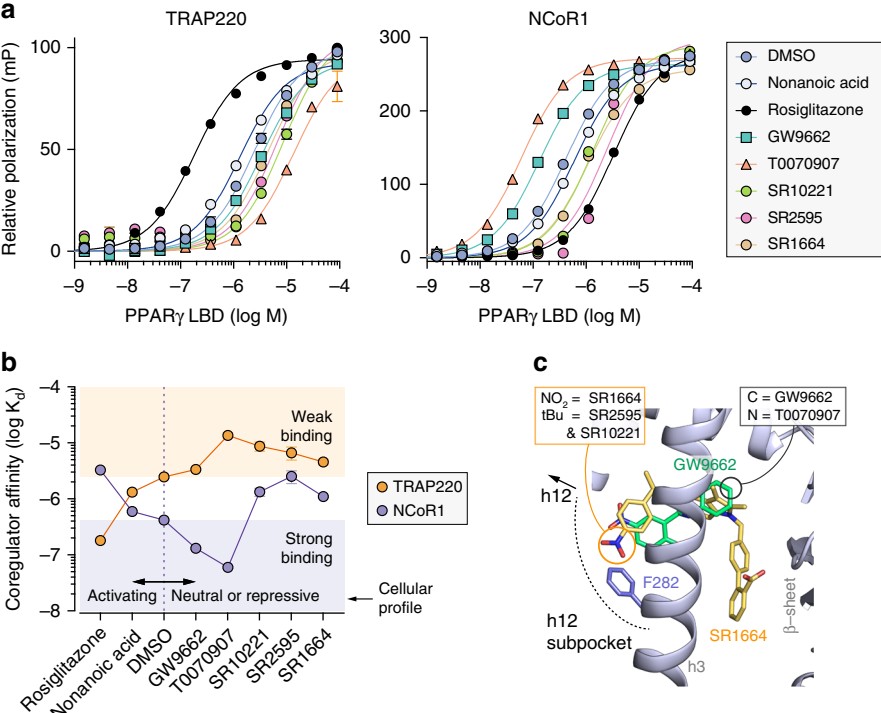

**Fig. 2** Coregulator binding profiles of the activating, neutral, and repressive synthetic PPARγ ligands. **a** Fluorescence polarization coregulator interaction assay showing the effect of ligands on the interaction between the PPARγ LBD and peptides derived from the TRAP220 coactivator or NCoR1 corepressor, plotted as the mean ($n = 2$) with error bars representative of s.d. **b** $K_d$ values derived from the coregulator interaction assay. Dotted orange and purple lines note the DMSO/apo-PPARγ values. Orange and purple shaded areas note the affinity regions for an ideal inverse agonist (i.e., for weaker TRAP220 affinity, orange circles in the orange square; for higher NCoR1 affinity, purple circles in purple squares). **c** Superposition of crystal structures of the PPARγ LBD bound to GW9662 (PDB code 3B0R; ligand in green, cartoon in blue) and SR1664 (PDB code 4R2U; ligand in yellow). The location of the simple substitution between GW9662 (methine) vs. T0070907 (nitrogen) is marked with a black circle, and the SR2595 and SR10221 tert-butyl extension within the helix 12 subpocket towards F282 (blue) from the SR1664 parent compound is marked with a yellow circle. Data are representative of at least 3 independent experiments

Table 1) and compared our structure to an available crystal structure of GW9662-bound PPARγ LBD (PDB code 3B0R). Strong electron density was observed for T0070907 in chain B and lower, less defined electron density in chain A. In both structures, PPARγ crystallized in the same space group and contained a dimer in the asymmetric unit with the expected α-helical sandwich fold (Fig. 3a). Structural superposition revealed nearly identical backbone conformations between the two structures (Cα r.m.s.d.: overall, 1.7 Å; chain A only, 1.34 Å; chain B only, 1.73 Å), revealing no overall structural changes that could account for the different pharmacological profiles of T0070907 and GW9662.

Focusing on the pyridyl ring of T0070907, we observed a water-mediated hydrogen bond network connecting the pyridyl nitrogen to the Nε atom in the R288 side-chain (Fig. 3b). Furthermore, the guanidinyl side chain of R288 forms a bipartite hydrogen bond with the side chain of E295. In contrast, the hydrogen bond network in the GW9662-bound structure is not extensive due the lack of a hydrogen bond acceptor in the phenyl ring of GW9662 (Fig. 3c). In support of our crystal structure, we observed a nuclear Overhauser effect (NOE) signal at ~4.77 p.p.m., consistent with a water interaction, to the R288 Hε-Nε group (Supplementary Figure 8) in a 3D $^{15}$N-NOESY-HSQC NMR data for T0070907-bound PPARγ LBD (Fig. 3d). We confirmed this result using a Phase-Modulated CLEAN chemical EXchange (CLEANEX-PM) NMR experiment (Supplementary Figure 9), which detects water–protein interactions[28].

The R288 Hε-Nε NMR crosspeak is not observed for GW9662-bound PPARγ (Supplementary Figure 8) likely due to chemical

exchange on the intermediate NMR time scale ($k_{ex} \approx |\Delta \nu|$), indicating the pyridyl-water interaction is important for stabilizing the dynamics of the R288 side chain. To confirm the stability of the hydrogen bond network observed in the crystal structures, we performed molecular dynamics simulations of T0070907- and GW9662-bound PPARγ ranging from 4–26 microseconds in length (Fig. 3e) in structures simulated with the observed crystallized waters (xtal), as well as a model generated of T0070907-bound PPARγ from the GW9662-bound PPARγ crystal structure that was independently solvated without the crystallized waters (model). In the simulations, the pyridyl group of T0070907 was hydrogen bonded to a water molecule for a significant fraction of the simulation (65–95%), as was the water-bridged R288-T0070907 pyridyl (5–46%). In contrast, a direct interaction between R288 and the pyridyl group of T0070907 not mediated by water was lowly populated (<2%). A direct (4–64%) and water-bridged (3–61%) R288-E295 interaction was also confirmed. The extensive pyridyl-based water-mediated hydrogen bond network is not possible to the hydrophobic phenyl group of GW9662, revealing a unique chemical feature in T0070907 that could confer corepressor-selective inverse agonism.

**The pyridyl-water network is essential for inverse agonism**. To test the functional role of the pyridyl-water hydrogen bond network observed in our T0070907-bound crystal structure, we generated variants of PPARγ by mutating residues that we predicted would maintain or break the pyridyl-water network. We hypothesized that a different positively charged residue (R288K)

**Table 1 X-ray crystallography data collection and refinement statistics**

|  | T0070907-bound PPARγ LBD |
|---|---|
| *Data collection* | |
| Space group | C 1 2 1 |
| *Cell dimensions* | |
| *a, b, c* (Å) | 92.99, 61.69, 118.66 |
| *α, β, γ* (°) | 90, 102.31, 90 |
| Resolution (Å) | 45.43–2.26 (2.341–2.26)[a] |
| $R_{sym}$ or $R_{merge}$ | 0.02979 (0.3361) |
| $I/\sigma I$ | 10.73 (2.12) |
| Completeness (%) | 99.53 (99.84) |
| Redundancy | 2.0 (2.0) |
| *Refinement* | |
| Resolution (Å) | 2.26 |
| No. reflections | 61110 |
| $R_{work}/R_{free}$ | 21.54/28.38 |
| *No. atoms* | |
| Protein/covalent ligand | 4181 |
| Water | 308 |
| *B-factors* | |
| Protein | 32.35 |
| Ligand/ion | 44.64 |
| Water | 30.64 |
| *R.m.s. deviations* | |
| Bond lengths (Å) | 0.008 |
| Bond angles (°) | 0.94 |

[a]Values in parentheses are for highest-resolution shell

would maintain the pyridyl-water network, whereas a hydrophobic residue (R288A or R288L) would break the pyridyl-water network. If the hydrogen bond network is important for corepressor-selective inverse agonism, we hypothesized that breaking this network via hydrophobic R288 mutations would afford a similar functional efficacy profile for both T0070907 and GW9662. We also generated a E295A mutation to test the importance of the bipartite hydrogen bond between R288 and E295. These mutations did not affect the structural integrity or stability of the PPARγ LBD as assessed by circular dichroism (CD) spectroscopy (Supplementary Figure 10).

Using a cell-based transcription assay, we tested the combined effect of the mutants and covalent ligands on PPARγ cellular activation (Fig. 4a). Wild-type PPARγ and the R288K mutant showed a similar profile, where cells treated with T0070907 showed decreased PPARγ transcription compared to DMSO or GW9662 treatment. In contrast, T0070907 did not decrease transcription for the R288A or R288L mutants, indicating the R288-mediated pyridyl-water network is responsible for the transcriptional repression conferred by T0070907. The E295A mutant maintained the cellular efficacy preference of T0070907 over GW9662, indicating that bipartite hydrogen bond is not a major contributor to the repressive activity, though both GW9662 and T0070907 showed lower activity compared to wild-type PPARγ.

We next tested the effect of the mutants on coregulator recruitment by determining binding affinities for the TRAP220 (Fig. 4b) and NCoR1 (Fig. 4c) peptides for wild-type PPARγ LBD and the mutant variants with or without pretreatment with GW9662 or T0070907. Consistent with the cell-based transcription assay, the R288K and E295A mutants maintained the coregulator binding profile of T0070907 and rank ordering. In contrast, the R288A and R288L mutants showed similar affinity for TRAP220 and NCoR1 when covalently bound to T0070907 or GW9662. This indicates the pyridyl-water network directs the

corepressor-selective inverse agonism profile of T0070907, the lack of which results in an antagonist GW9662-like profile.

To more robustly compare how the wild-type and mutant PPARγ variants performed in the above assays, we performed a version of the web of efficacy analysis used in the G-protein coupled receptor (GPCR) field to study ligand signaling bias[29,30]. We plotted the multivariate data on a radar chart with axes corresponding to each of the assays whereby conditions that are biased towards corepressor-selective inverse agonism populate the outer ring of the radar chart, and less favorable conditions populate the center (Fig. 4d). The analysis clearly shows that T0070907 selects corepressor-selective inverse agonism functions only for wild-type PPARγ and the R288K mutant variant (Fig. 4e). This dramatic result reveals that R288-mediated pyridyl-water network directs the corepressor-selective inverse agonism profile conferred by T0070907.

**T0070907-bound PPARγ populates two long-lived conformations.** In principle, there should be structural differences in the AF-2 coregulator interaction surface to account for the different pharmacological profiles of T0070907 and GW9662. However, in the crystal structures of PPARγ bound to GW9662 or T0070907 the conformation of the AF-2 surface is influenced by crystal contacts (Supplementary Figure 11). Chain A adopts an active conformation with helix 12 docked into the AF-2 surface; however, helix 12 in chain B is distorted due to crystal contacts and docks into the AF-2 surface of a neighboring symmetry related chain A molecule, thus influencing the conformation of helix 12 in both chains. Indeed, it is difficult to determine the structural mechanism of action of PPARγ ligands from crystal structures alone[31]. However, NMR studies have shown that the orthosteric pocket and helix 12 are dynamic and switch between two or more conformations on the microsecond-to-millisecond (μs-ms) time scale in the ligand-free/apo-form. This results in very broad or unobserved NMR peaks for residues in helix 12 and the orthosteric pocket, which are stabilized upon binding a full agonist[6,15,31]. We therefore used NMR spectroscopy to assess the impact of T0070907 and GW9662 on the dynamics of the PPARγ LBD.

NMR data of apo-PPARγ are similar to PPARγ covalently bound to GW9662 with widespread μs-ms dynamics in the ligand-binding pocket and helix 12/AF-2 surface[32]. In contrast, T0070907-bound PPARγ showed a remarkably widespread stabilization of μs-ms dynamics, evident by the appearance of peaks in 2D [$^1$H,$^{15}$N]-transverse relaxation optimized spectroscopy-heteronuclear single quantum coherence (TROSY-HSQC) NMR spectrum (Fig. 5a). This includes well-resolved T0070907-bound NMR peaks corresponding to residues located in close proximity to the T0070907 R288-mediated pyridyl-water network (Fig. 5b) in the β-sheet (V248, G344, G346), helix 3 (I279, G284) and the adjacent helix 7 (G361). Furthermore, a peak for V322 on helix 5 within the AF-2 surface appeared, indicating stabilization of the AF-2 surface. The T0070907-bound crystal structure also shows a larger network of water-mediated hydrogen bonds not involving R288 that form a molecular hub linking the pyridyl-water network to the β-sheet (via backbone hydrogen bonds to I341 and E343) and helix 5 (via backbone hydrogen bond to I326).

In temperature-dependent NMR studies (Fig. 5c), we observed persistent NMR peak doubling for a number of PPARγ residues when bound to T0070907 but not GW9662, including G399 located near the AF-2 surface (Fig. 5b). Peak doubling indicates the presence of two long-lived T0070907-bound structural conformations in slow exchange on the NMR time scale, where the difference in chemical shift between the two states ($\Delta\nu$, in Hz)

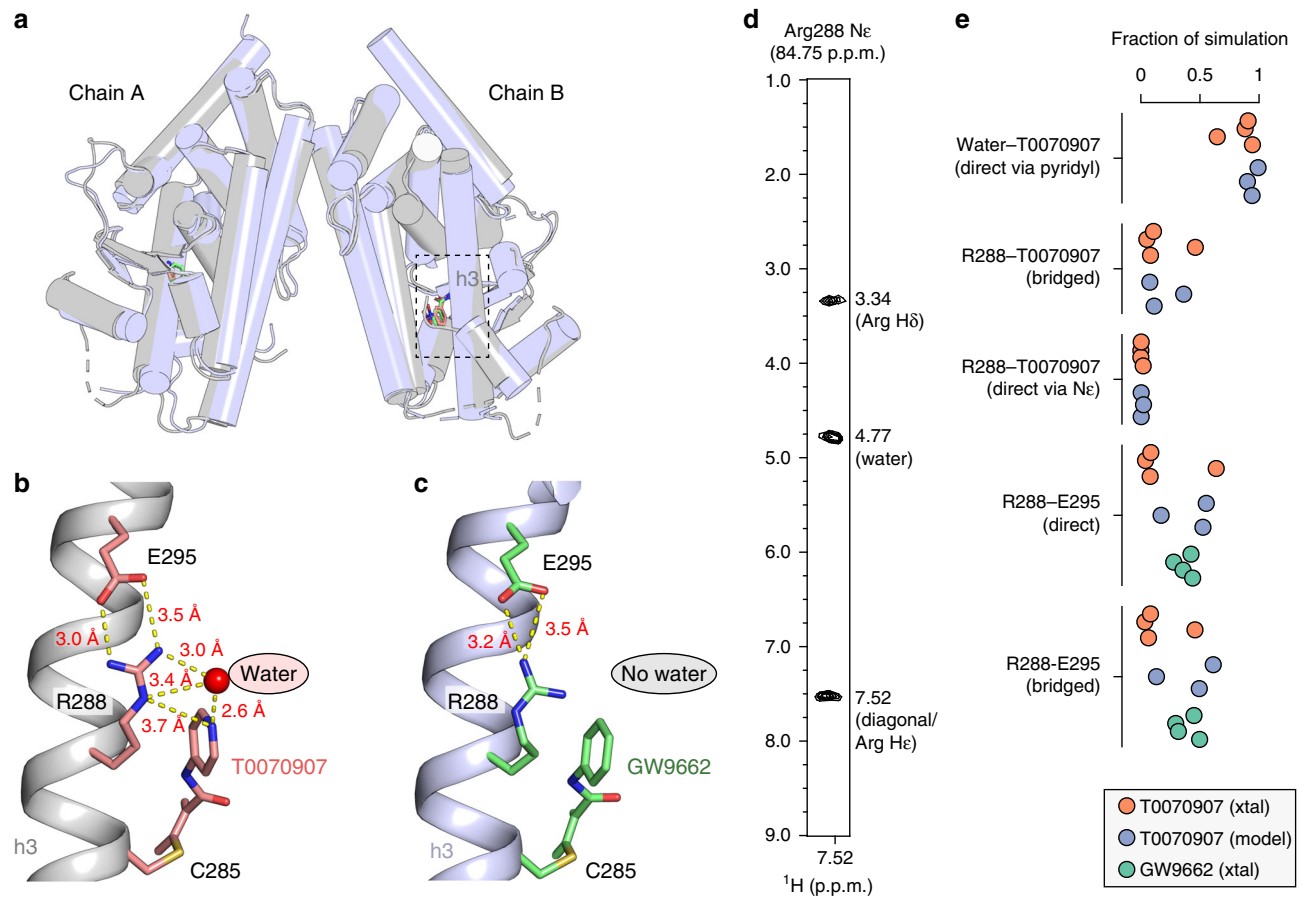

**Fig. 3** A water-mediated hydrogen bond network connects Arg288 to T0070907. **a** Overall structure of T0070907-bound PPARγ LD (PDB code 6C1I; gray) and overlay with GW9662-bound crystal structure (PDB code 3B0R; blue). **b** A water-mediated hydrogen bond network in the T0070907-bound crystal structure (chain B is shown) links the pyridyl group in T0070907 to the R288 side chain, which forms a bipartite hydrogen bond with the E295 side chain. **c** The GW9662-bound crystal structure (chain B is shown) lacks the R288-ligand hydrogen bond network but contains the R288-E295 hydrogen bond. **d** 2D strip corresponding to the R288 Nε-Hε group from a 3D $^{15}$N-NOESY-HSQC NMR experiment ($\tau_{mix} = 100$ ms) of T0070907-bound $^{15}$N-labeled PPARγ LBD reveals a NOE signal at 4.77 p.p.m. confirming a direct water interaction. **e** Pyridyl-water network hydrogen bonds populated during molecular dynamics simulations for T0070907- and GW9662-bound structures starting from crystallized (xtal) and modeled (model) conformations from 3–4 replicate simulations

is much greater than the exchange rate ($k_{ex}$) between conformations on the order of milliseconds-to-seconds (ms–s)[33]. ZZ-exchange NMR experiments (also called EXSY, or exchange spectroscopy) enable detection of the interconversion between long-lived structural states via transfer of the $^1$H chemical shift of one state to the other when $k_{ex} \approx 0.2-100\ s^{-1}$ and $k_{ex} >> \Delta v$. Exchange crosspeaks for G399, which showed well-dispersed peak doubling, were observed at 310 K but not at 298 K (Fig. 5d), indicating the exchange between the two conformations is too slow to be measured at room temperature ($k_{ex} < 0.2\ s^{-1}$). To determine an exchange rate, we performed ZZ-exchange experiments with varying exchange delays at 310 K and fit the data to a two-state interconversion model (Fig. 5e), which provided an exchange rate of ~2.1 s$^{-1}$ between the upfield-shifted state ($P_A = 37\%$; $k_{A \to B} = 0.8\ s^{-1}$) and downfield-shifted state ($P_B = 63\%$; $k_{B \to A} = 1.3\ s^{-1}$). The widespread peak doubling (Supplementary Figure 12A) indicates a global conformational change, which is slow on the NMR time scale. In most cases, spectral overlap did not permit fitting of the data to extract an exchange rate. However, residues with notable peak doubling comprise distant structural regions that are also connected via the aforementioned extended pyridyl-water network (Fig. 5b), including the β-sheet (G338) and helix 6 (R350, S355, L356) within the ligand-binding

pocket; a surface comprising helix 2a (R234) and the C-terminal region of helix 7 and the loop connecting helix 7 and 8 (K373, N375, E378, D380); helix 3 near the AF-2 surface (I303); and the loop connecting helix 8 and 9 near the AF-2 surface (S394), including G399.

Our NMR analysis revealed that T0070907-bound PPARγ undergoes a global conformational change between two long-lived structural conformations. The conservation of peak doubling at higher temperatures indicates both conformations are stable on a timescale greater than 1 ms. The line broadening observed and/or absence of peaks at lower temperature for some doubled sets of peaks (e.g., T238, G321, G346) indicates that within each of the two long-lived conformation(s) populated by T0070907 there is intermediate exchange between two or more conformations on the NMR timescale. Thus, the overall increase in NMR peaks when PPARγ is bound to T0070907 is due to the presence of two long-lived structural conformations, or tier 0 conformations[34], and stabilization of μs-ms dynamics of individual conformations within each long-lived conformation (Fig. 5f). Interestingly, the peak for G321 in the GW9662-bound conformation and one of the T0070907-bound conformations shows significant line broadening at the lower temperature, indicating these states may share conformational and dynamical features.

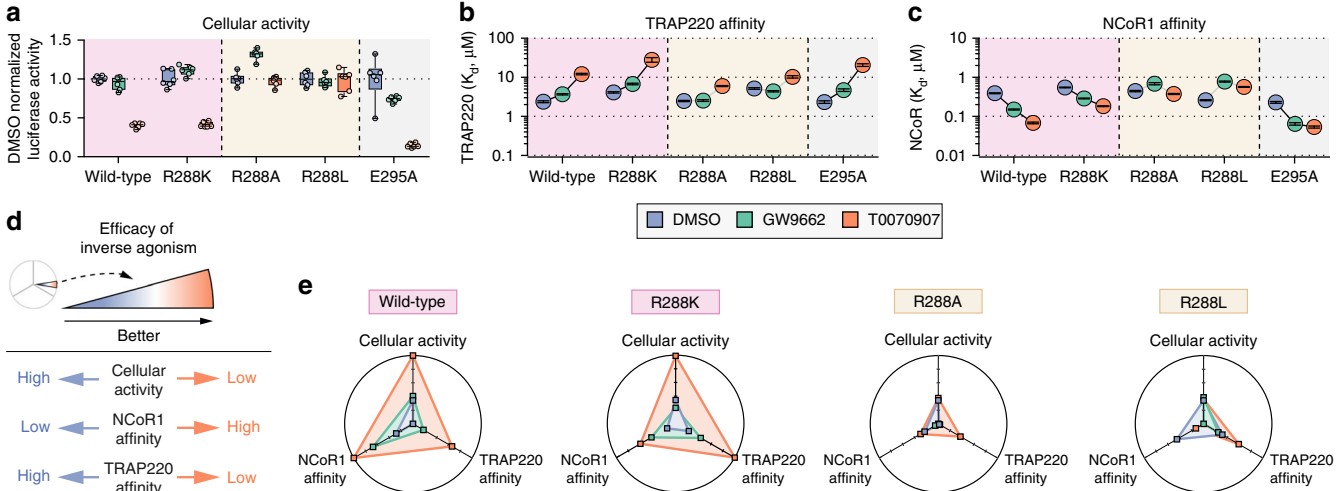

**Fig. 4** Mutagenesis validates the R288-pyridyl interaction for conferring corepressor-selective inverse agonism. **a** HEK293T cells transfected with full-length PPARγ expression plasmid along with a 3 × -PPRE-luciferase reporter plasmid and treated with the indicated ligands (5 μM). Individual points ($n = 6$) normalized to DMSO control (mean) are plotted on top of a box-and-whiskers plot; the box represents 25th, median, and 75th percentile of the data, and the whiskers plot the entire range of values. **b**, **c** Affinities determined from a fluorescence polarization assay of wild-type and mutant PPARγ LBDs preincubated with a covalent ligand (GW9662 or T0070907) or vehicle (DMSO) binding to FITC-labeled **b** TRAP220 or **c** NCoR1. Data plotted as the $K_d$ value and error from fitting data of two experimental replicates using a one site binding equation. **d** Legend to the web of efficacy radar chart diagrams. Corepressor-selective inverse agonism is associated with data points populating the periphery of the plots. **e** Radar web of efficacy plots displaying assay data for wild-type PPARγ and mutant variants. Data normalized within the range of values for each assay (**a–c**). Data are representative of at least 3 independent experiments

**A mutual conformation with GW9662 and a unique conformation**. G399 is an ideal NMR observable probe that is sensitive to the conformation of the AF-2 surface: it is linked to the AF-2 surface through water-mediated hydrogen bonds to N312 and D313 on helix 5 but does not directly interact with a bound coregulator peptide (Fig. 6a). Strikingly, for G399 and the other residues that showed peak doubling in the ZZ exchange analysis, we found that the backbone amide chemical shifts of one of the two peaks observed for T0070907-bound PPARγ are similar to the single peak observed for GW9662-bound PPARγ (Fig. 6b and Supplementary Figure 12B). This indicates that one of the two long-lived T0070907-bound conformations is structurally similar to GW9662-bound PPARγ (mutual conformation) and the other is uniquely populated only when bound to T0070907 (unique conformation). Consistent with the mutant activity data (Fig. 4), NMR analysis revealed that T0070907-bound R288K mutant protein significantly populated the unique (state B) conformation (Fig. 6b and Supplementary Figure 12C). In contrast, T0070907-bound R288A mutant protein, which showed an antagonist profile similar to GW9662, significantly populated the GW9662-bound mutual (state A) conformation (Fig. 6b and Supplementary Figure 12D). A lowly populated unique conformation is also observed for the T0070907-bound R288A mutant, which indicates the aforementioned extended pyridyl-water hydrogen bond network observed in the crystal structure lowly populates the unique conformation but is not sufficient for directing corepressor-selective inverse agonism. However, the pyridyl-water interaction with R288, or a positively charged residue (e.g., R288K), is necessary for significant population of the unique conformation and directing corepressor-selective inverse agonism.

We also assessed the conformational state of helix 12 directly using ¹⁹F NMR (Fig. 5c) by attaching the ¹⁹F NMR-detectible probe 3-bromo-1,1,1-trifluoroacetone (BTFA) on K474 (Fig. 5a). Consistent with our previous result[14], the ¹⁹F spectral profile of GW9662-bound PPARγ revealed two peaks corresponding to a major state (right peak; 78%) and minor state (left peak; 22%).

T0070907-bound PPARγ also showed two peaks with chemical shift values similar to GW9662-bound PPARγ, but the population magnitudes of the states are switched and skewed towards the left peak. Strikingly, this helix 12/AF-2 surface ¹⁹F NMR probe showed similar relative population sizes to that observed in the G399 ZZ-exchange analysis (34% and 66%, respectively). The right ¹⁹F NMR peak abundantly populated by GW9662 and moderately populated by T0070907 likely corresponds to the mutual G399 conformation from the 2D NMR analysis. In contrast, the left ¹⁹F NMR peak likely corresponds to the unique G399 conformation; this peak is abundantly populated by T0070907 but lowly populated by GW9662. The low abundance of this peak when bound to GW9662 could explain in part why it was not detected by the 2D NMR analysis, which has lower overall sensitivity of signal-to-noise compared to the ¹⁹F NMR analysis. However, the BTFA probe attached to helix 12 may also be more sensitive to larger structural changes compared to backbone amide resonances.

**T0070907 prepopulates a corepressor-bound conformation**. To determine whether the unique and mutual long-lived T0070907-bound conformations would display similar or distinct coregulator interaction preferences, we titrated the NCoR1 corepressor and TRAP220 coactivator peptides and monitored their binding to T0070907-bound ¹⁵N-labeled PPARγ LBD by NMR. Remarkably, titration of NCoR1 peptide (Fig. 7a, b) resulted in a decrease of the mutual G399 conformation (state A) associated with an increase and slightly shifting of the unique G399 conformation (state B) towards a peak with similar chemical shift values and intensity to the unique conformation. The shifting of the unique conformation occurs before the mutual conformation shifting completes, indicating that NCoR1 binds with higher affinity to the unique conformation. Using NMR line shape modeling[35] considering the exchange rates and molar fractions of the two slowly exchanging populations, we found that the overlay of 1D ¹⁵N planes extracted from the 2D NMR data (Fig. 7c) are best fit (Fig. 7d) using a 4-state model (Supplementary Figure 13)

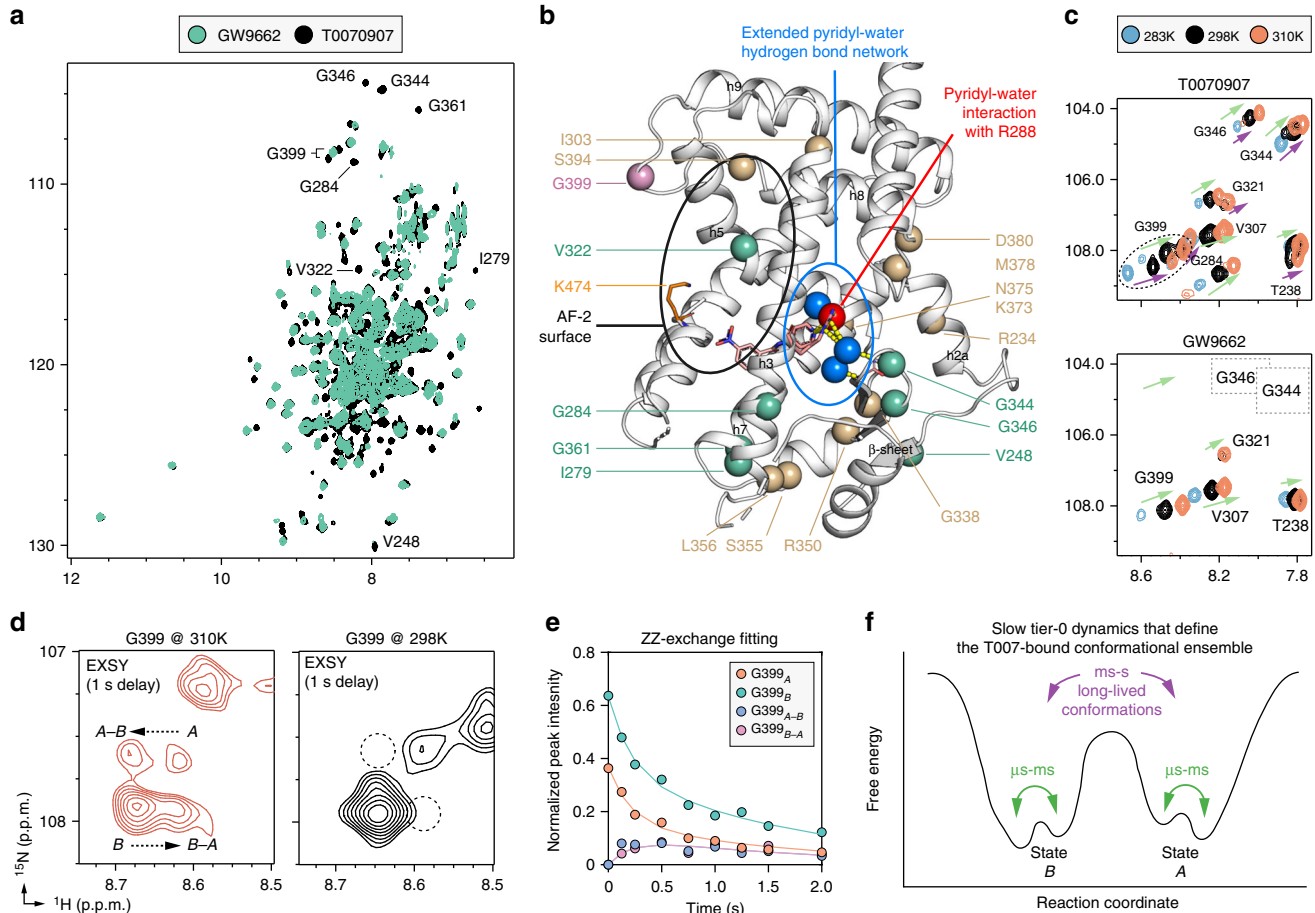

**Fig. 5** NMR detected exchange between two long-lived T0070907-bound conformations. **a** Overlay of 2D [$^1$H,$^{15}$N]-TROSY-HSQC NMR spectra of $^{15}$N-labeled PPARγ LBD bound to GW9662 or T0070907. **b** Binding of T0070907 but not GW9662 stabilizes intermediate exchange (μs-ms time scale) dynamics (residues labeled in **a** shown in green spheres) and causes peak doubling (tan and pink spheres; G399 is colored pink for emphasis). Data plotted on the T0070907-bound PPARγ crystal structure and important structural regions are highlighted as follows: AF-2 surface (black oval); an extended pyridyl-water hydrogen bond network (blue spheres, yellow dotted lines, blue oval), beyond the key pyridyl-water interaction (red sphere). **c** Snapshot overlays of 2D [$^1$H,$^{15}$N]-TROSY-HSQC spectra of $^{15}$N-labeled PPARγ LBD bound to T0070907 or GW9662 shows co-linear shifting of peaks at the different temperatures. The spectral region displayed shows peaks conserved when PPARγ is bound to GW9662 or T0070907 (green arrows); a unique set of doubled peaks when bound to T0070907 (purple arrows); and absent peaks due to intermediate exchange on the NMR time scale when bound to GW9662 (dotted rectangles). **d** Snapshots of ZZ-exchange [$^1$H,$^{15}$N]-HSQC NMR spectra (delay = 1 s) of T0070907-bound $^{15}$N-labeled PPARγ LBD focused on G399 at the indicated temperatures. Two G399 conformational states are denoted as A and B with the ZZ-exchange transfer crosspeaks as A–B and B–A. **e** ZZ-exchange NMR analysis build-up curve from for G399 at 310 K generated by plotting peak intensities of the state A and B peaks and exchange crosspeaks (A–B and B-A) as a function of delay time. **f** Schematic of the T0070907-bound PPARγ conformational ensemble defined by the NMR studies

where the two slowly exchanging (i.e., isomerization) receptor populations (R and R*) are capable of binding to the peptide (RL and R*L). This model also accounts for receptor isomerization in peptide-bound states (R ↔ R*, and RL ↔ R*L), and the fitting indicates that lower affinity NCoR1 binding to the mutual state (state A) results in a slow conformational change (isomerization) to the final NCoR1-bound state. This latter point is apparent in the NMR titration data because as the mutual conformation (state A) disappears upon NCoR1 binding, it shifts away rather than towards the unique conformation (state B) and NCoR1-bound state. This suggests an induced fit binding mechanism by which NCoR1 binding to the mutual conformation results in a conformational change to a more thermodynamically stable species. We next examined the effect of TRAP220 peptide binding to T0070907-bound $^{15}$N-labeled PPARγ LBD. In contrast to the NCoR1 results, titration of TRAP220 peptide (Fig. 7e, f) resulted in a decrease of the unique G399 conformation (state B) followed by an increase and shifting of the mutual G399 conformation

(state A) towards a peak with similar chemical shift values and intensity as the mutual conformation. Analogous but opposite to the NCoR1 titration, the shifting of the mutual conformation occurs before the unique conformation shifting completes, indicating that TRAP220 binds with higher affinity to the unique conformation. Because the equilibrium binding affinity of TRAP220 is weaker than NCoR1 for T0070907-bound PPARγ, more TRAP220 peptide was required to saturate the observed NMR changes (Supplementary Figure 14). NMR line shape modeling again revealed that the overlay of 1D $^{15}$N planes extracted from the 2D NMR data (Fig. 7g) were best fit (Fig. 7h) using a 4-state model (Supplementary Figure 13) accounting for receptor isomerization in the peptide-free and peptide-bound states. Analogous to the NCoR1 titration, the unique conformation shifted away from rather than towards the TRAP220-bound state suggesting an induced fit binding mechanism. Notably, the same coregulator binding trends for the other T0070907-bound PPARγ residues with peak doubling, where NCoR1 binding shifts

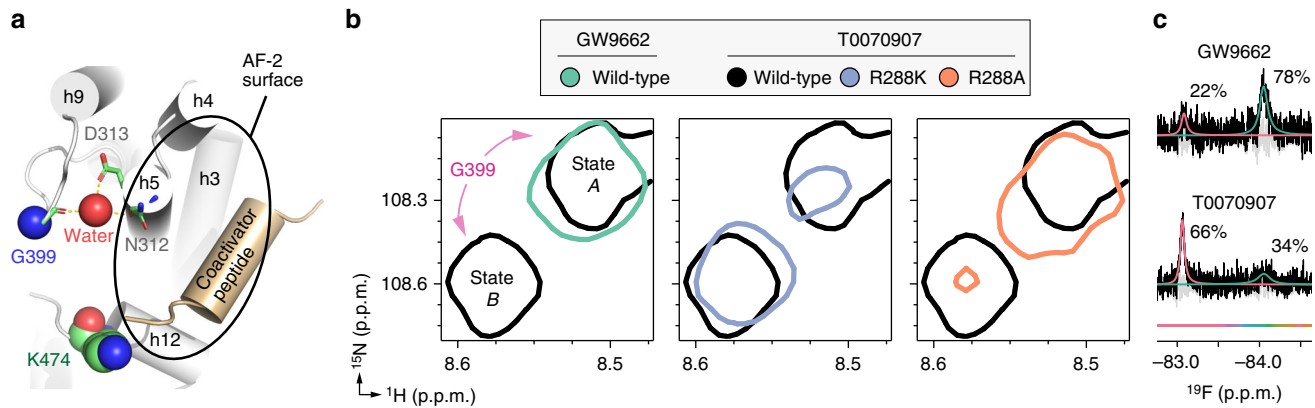

**Fig. 6** T0070907 populates a shared conformation with GW9662 and a unique conformation. **a** Structural location of G399, which is connected to the AF-2 coregulator interaction surface via water-mediated hydrogen bonds to N312 and D313 but does not directly interact with a coregulator peptide bound to the PPARγ LBD (PDB code 2PRG). **b** Snapshot overlay of [¹H,¹⁵N]-TROSY-HSQC NMR spectra of ¹⁵N-labeled PPARγ LBD (wild-type or R288 mutants) bound to GW9662 or T0070907 shows that the single GW9662-bound G399 peak has similar chemical shift values to one of the two T0070907-bound G399 peaks (state A) whereas state B is uniquely populated by T0070907. The T0070907-bound R288K mutant PPARγ LBD, but not T0070907-bound R288A mutant PPARγ LBD, significantly populates the unique conformation. **c** Deconvoluted ¹⁹F NMR spectra of PPARγ LBD labeled with BTFA on helix 12 and covalently bound to GW9662 or T0070907

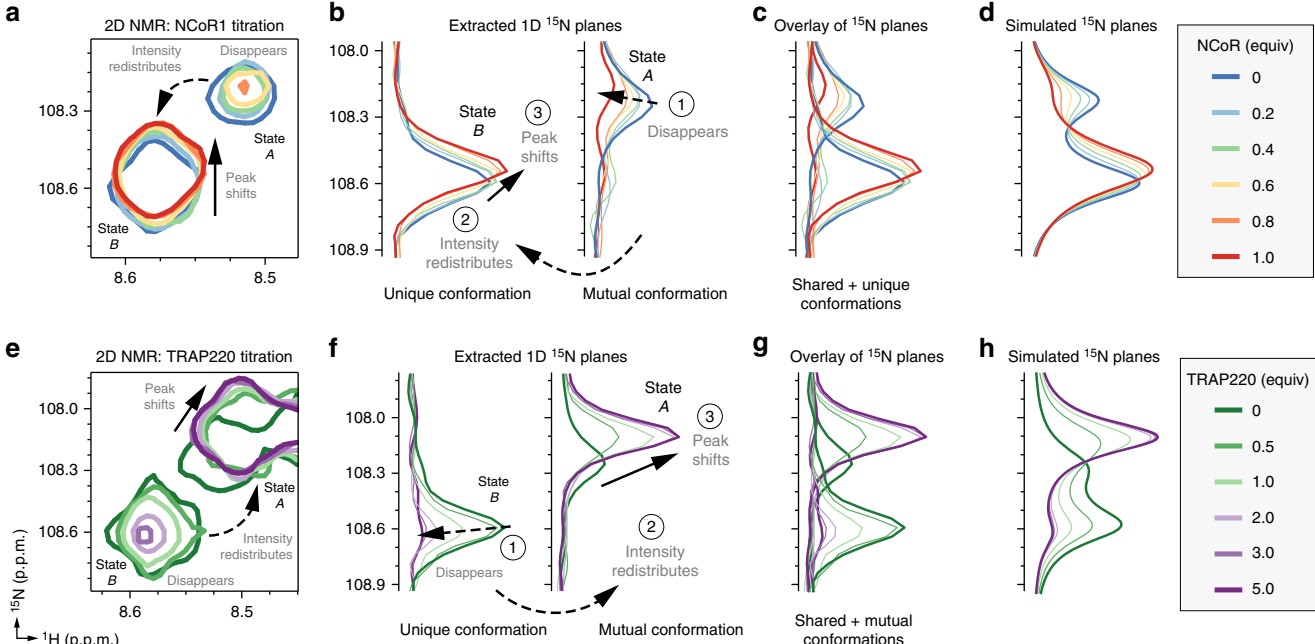

**Fig. 7** Coregulator binding preferences of the unique and mutual T0070907-bound conformations. **a** Snapshot overlay of 2D [¹H,¹⁵N]-TROSY-HSQC spectra of T0070907-bound ¹⁵N-labeled PPARγ LBD titrated with NCoR1 peptide. **b** Extracted 1D planes of the NCoR1 spectra shown in **a** show the qualitative binding trends observed by NMR, whereas **c** an overlay of the extracted 1D planes is quantitatively described by **d** spectra using a 4-state model where the two slowly exchanging receptor populations (R and R*) are capable of binding to the peptide (RP and R*P), and receptor isomerizes in both the free and peptide-bound states (R ↔ R*, and RP ↔ R*P). (**e–h**) Analysis of performed for T0070907-bound ¹⁵N-labeled PPARγ LBD titrated with TRAP220 peptide performed similarly to the NCoR1 analysis described in **a–d**, which is also quantitatively described using the same 4-state model

the peak populations towards the unique state B (Supplementary Figure 12D) and TRAP220 binding shifts the peak populations towards the mutual state A (Supplementary Figure 12E).

We also used NMR to examine the coregulator binding properties of GW9662-bound ¹⁵N-labeled PPARγ LBD, which only shows the mutual conformation. Interestingly, whereas NCoR1 or TRAP200 binding to T0070907-bound PPARγ consolidated the unique and mutual conformations into one coregulator-bound conformation, NCoR1 binding to GW9662-

bound PPARγ LBD caused peak doubling of the GW9662-bound G399 NMR peak towards chemical shift values similar to the NCoR1- and TRAP220-bound forms of T0070907-bound PPARγ LBD (Fig. 8a). NMR line shape modeling revealed that the 2D NMR profile was best fit (Fig. 8b) using a 3-state model (Supplementary Figure 13) where the receptor binds to the peptide (R + P ↔ RP), and the peptide-bound receptor slowly isomerizes between two states (RP ↔ R*P). In contrast, TRAP220 binding only shifted the single GW9662-bound G399 peak

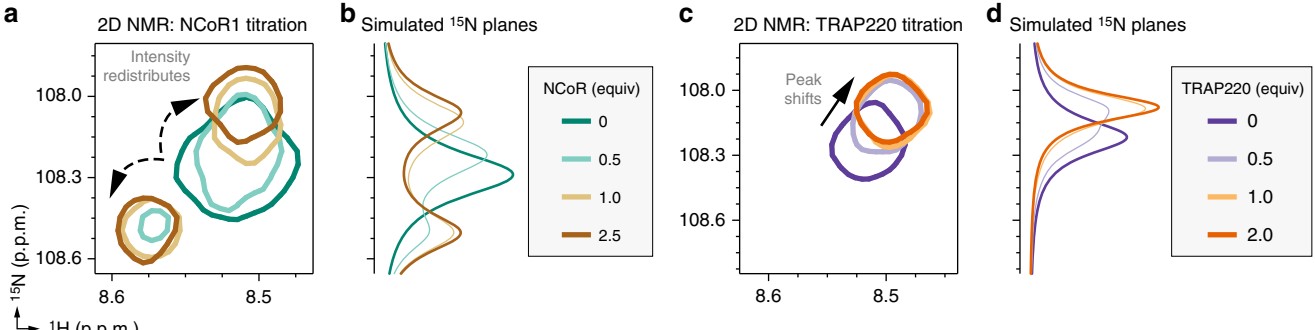

**Fig. 8** Coregulator binding preferences of the single GW9662-bound conformation detected by NMR. **a** Snapshot overlay of 2D [$^1$H,$^{15}$N]-TROSY-HSQC spectra of GW9662-bound $^{15}$N-labeled PPARγ LBD titrated with NCoR1 peptide. **b** The $^{15}$N spectral planes of the 2D NMR titration are quantitatively described using a 3-state model (R + P ↔ RP ↔ R*P) where PPARγ LBD bound to GW9662 and NCoR1 slowly isomerizes between two states (RP and R*P). **c** Snapshot overlay of 2D [$^1$H,$^{15}$N]-TROSY-HSQC spectra of GW9662-bound $^{15}$N-labeled PPARγ LBD titrated with TRAP220 peptide. **d** The $^{15}$N spectral planes of the 2D NMR titration are quantitatively described using a 2-state model (R + P ↔ RP)

towards the TRAP220-bound form of T0070907-bound PPARγ (Fig. 8c), which was best fit (Fig. 8d) using a simple 2-state model (R + P ↔ RP) (Supplementary Figure 13).

These results reveal that the two long-lived T0070907-bound conformations have different binding preferences for NCoR1 and TRAP220. The NMR chemical shifts of the unique and mutual T0070907-bound conformations in the absence of coregulator peptide are similar to the NCoR1- and TRAP220-bound forms, respectively. Thus, the unique and mutual T0070907-bound states prepopulate a corepressor-like and coactivator-like bound conformation that afford high-affinity binding to NCoR1 and TRAP220, respectively. Furthermore, the chemical shift difference between the unique T0070907 conformation and NCoR1-bound state (i.e., the degree of state B shifting) is much smaller than the mutual conformation and TRAP220-bound state (i.e., the degree of state A shifting), indicating that the corepressor-like conformation prepopulated by T0070907 is more similar to the corepressor-bound state. In contrast, NCoR1 binding to GW9662-bound PPARγ introduces a conformational frustration within the AF-2 surface: the AF-2 surface of GW9662-bound PPARγ does not prepopulate the corepressor-bound conformation and upon binding NCoR1is found in two slowly exchanging conformations similar to the corepressor- and coactivator-bound forms of T0070907-bound PPARγ.

## Discussion

Carbon (methine)-to-nitrogen ligand substitutions are known to have beneficial effects on pharmacological parameters[36], though it is difficult to predict how subtle changes in chemical structure impact functional efficacy[37,38]. The original discoveries of GW9662 and T0070907 referred to these compounds as antagonists not because of their pharmacological properties, but because they bind covalently to the orthosteric ligand-binding pocket of PPARγ and physically block the binding of other ligands, thereby antagonizing PPARγ activation by orthosteric agonists[21,39]. Importantly, our analyses herein and other work[12,14,21,39] shows that GW9662 and T0070907 have unique biochemical and cellular transcriptional properties that are separate from their ability to block other ligands from binding to the orthosteric pocket, but not to an alternate ligand-binding site[32,40,41]. Thus, in addition to their utility as nondissociative orthosteric competitive ligands that inhibit binding of other orthosteric PPARγ ligands, our data suggest that T00709707 and GW9662 can be classified as a covalent corepressor-selective inverse agonist and a covalent antagonist, respectively.

Our studies illuminate a structural mechanism affording the corepressor-selective inverse agonism of PPARγ. Our crystal structure of T0070907-bound PPARγ revealed a water-mediated hydrogen bond network linking the critical corepressor-selective inverse agonist switch residue (R288) to the pyridyl group of T0070907. Our NMR analysis shows that T0070907-bound PPARγ, but not GW9662-bound PPARγ, slowly exchanges between two long-lived conformations. One of these conformations is shared with GW9662-bound PPARγ, significantly populated by the R288A mutant that cannot form the pyridyl-water-R288 network and shows similar backbone amide chemical shift values indicating a similar conformation to the coactivator-bound state. The other conformation is uniquely and abundantly populated by T0070907, significantly populated by the R288K mutant, and highly similar to the corepressor-bound state, affording higher affinity corepressor binding and transcriptional repression. This indicates the conformation of the PPARγ AF-2 surface is primed for high affinity binding to NCoR1 when PPARγ is bound to T0070907. Notably, given that the NMR-detected exchange rate between the two T0070907-bound conformations is greater than 1 s, access to helix 12/AF-2 conformations consistent with our NMR data in molecular dynamics simulations would be inaccessible with current standard simulation approaches. However, accelerated molecular dynamics and metadynamics simulations have revealed that helix 12 in PPARγ can exist in two conformations with similar free energy[42] and recent structural studies have captured non-active helix 12 conformations[43,44]; these conformations may be similar to the unique long-lived T0070907-bound AF-2 conformation that we observed by NMR that primes PPARγ for high affinity corepressor binding. Overall, our work shows that the combination of different but complementary structural methods provides the full picture of ligand mechanism of action.

The notion that T0070907 can repress PPARγ-mediated transcription raises the question as to whether or not PPARγ is constitutively active. PPARγ is activated by endogenous fatty acids and lipids, so the repressive effects of synthetic ligands could be due to displacement of activating endogenous ligands by ligands that do not activate PPARγ transcription to the same degree. However, four lines of evidence indicate that PPARγ does possess intrinsic activity in the absence of a bound ligand. First, a PPARγ mutant (Q286P) unable to be activated by synthetic ligands or natural ligands showed an increase in transcription to a similar degree as wild-type PPARγ relative to cells transfected with an empty control plasmid[24]. Transduction of PPARγ-null fibroblasts with this mutant promoted adipogenesis and

generated fat pads when the fibroblasts were injected into mice. These findings indicate that ligand binding is not required for PPARγ-mediated adipocyte differentiation. Second, structural evidence for constitutive PPARγ activity is suggested by a crystal structure of apo-PPARγ LBD (PDB code 3PRG), in which the conformation of helix 12 is found in the active conformation without any bound ligand or coregulator peptide[45]. Importantly, the conformation of helix 12 does not appear to be influenced by crystal contacts as is the case in our T0070907-bound structure or the GW9662-bound structure (PDB code 3B0R). Third, we showed that transfection of PPARγ into cells cultured in charcoal stripped FBS shows increased luciferase activity relative to empty control plasmid, similar to normal FBS, and the activating and repressive ligand activity profiles were the same in normal FBS and charcoal stripped FBS. Although this indicates that the components of the cell culture media do not have a significant influence on the pharmacological activities of the activating and repressive synthetic ligands tested, it is not possible to rule out other cellular ligands that may be produced within the cells unrelated to the cell culture media components. Fourth, if a cellular lipid or fatty acid were required for PPARγ activity, then apo-PPARγ LBD should not show meaningful affinity for cor-egulators. However, our coregulator profiling shows that the apo-PPARγ LBD interacts with peptides derived from coregulators[25,26]. Published findings and the data presented here indicate that PPARγ possesses constitutive activity and that binding of ligands differentially influences PPARγ activity. However, if indeed PPARγ lacks constitutive activity, our results should only require a minor change in nomenclature, as the corepressor-selective activity of T0070907 is well supported by structural and functional evidence provided by us and by others.

Our findings suggest a means for pharmacologically directing transcriptional repression via corepressor-selective inverse agonism of PPARγ. The previous finding that ligand engagement via hydrogen binding to helix 12 via Y473 is critical for mediating agonism transformed the way that PPARγ agonists were developed[46]. The AF-2 steric clash structural mechanism of action for the repressive inverse agonist PPARγ compounds SR2595 and SR10221[6,20] seems to employ a coregulator inhibition profile. In contrast, our studies here indicate that ligand hydrogen bonding to the guanidinyl side chain of R288, water-mediated or perhaps directly, may be a critical mediator of corepressor-selective PPARγ inverse agonism. Repressive PPARγ modulators show promise for improving the therapeutic index associated with anti-diabetic PPARγ ligands by promoting bone formation rather than decreasing bone mass[5,6], which occurs with agonists used clinically such as the TZDs. Furthermore, repression of PPARγ activity affects fat mobilization and may be a means to therapeutically treat obesity and extend lifespan[7], and T0070907 has demonstrated efficacy in cancer models[9–12]. Thus, our findings should inspire future work to develop and characterize corepressor-selective inverse agonists to probe the repressive functions of PPARγ.

## Methods

**Materials and reagents**. Human PPARγ LBD (residues 203–477 in isoform 1 numbering, which is commonly used in published structural studies and thus throughout this manuscript; or residues 231–505 in isoform 2 numbering), mutant PPARγ LBD proteins, full-length PPARγ (isoform 2), and full-length retinoid x receptor α (RXRα) were expressed in *Escherichia coli* BL21(DE3) cells using enriched media (LB or autoinduction) or for NMR studies minimal media (M9 supplemented with $^{13}$C-glucose and/or $^{15}$NH$_4$Cl) as TEV-cleavable hex-ahistidine-tagged fusion protein using a pET46 plasmid. Following expression, cells were lysed in lysis buffer (500 mM potassium chloride, 40 mM potassium phosphate, 15 mM imidazole, pH 7.5) supplemented with 5 μg/mL of DnaseI and lysozyme. Lysates were cleared by sonication (24,000 × $g$, 1 h) and loaded onto 2 × 5 mL Histrap FF columns (GE Healthcare). Protein was eluted using lysis buffer with 500 mM imidazole. For TEV cleavage, protein was incubated at a 1:50 ratio

with TEV protease overnight at 4 °C, loaded back onto the HisTrap FF column and collecting the flow through. Protein was concentrated and loaded onto a Superdex 200 prep grade 26/60 column (GE Healthcare). The final storage buffer for LBD samples following size exclusion chromatography and subsequently frozen at −80 °C was 50 mM potassium chloride (pH 7.4), 20 mM potassium phosphate, 5 mM TCEP, and 0.5 mM EDTA; for full-length PPARγ was 25 mM MOPS (pH 7.4), 300 mM potassium chloride, and 1 mM EDTA; or for full-length RXRα ws 25 mM MOPS (pH 7.4), 300 mM KCl, 1 mM EDTA, and 5 mM TCEP.

Synthetic ligands GW9662, T0070907, and SR1664 were obtained from Cayman Chemical; rosiglitazone was obtained from Tocris Bioscience and Cayman Chemical; nonanoic acid was obtained from Sigma; and SR2595 and SR10221 were previously synthesized in house[6]. In most studies of covalent ligands (except TR-FRET studies), PPARγ protein was pretreated with GW9662 or T0070907 overnight at 4 °C with a 2X molar excess of compound dissolved in d$_6$-DMSO. Delipidation was performed using Lipidex 1000 resin (Perkin-Elmer): the protein was diluted to 0.8 mg ml$^{−1}$, batched with an identical volume of resin at 37 °C and spun at 100 rpm for 45 min, drawn through a gravity column by syringe, and concentrated to a working concentration in 25 mM MOPS (pH 7.4), 25 mM potassium chloride, 1 mM EDTA, and was either frozen immediately at −80 °C or used in the same day. Mammalian expression plasmids included Gal4-PPARγ-hinge-LBD (residues 185-477 in isoform 1 numbering; 213-505 in isoform 2 numbering) inserted in pBIND plasmid; and full-length PPARγ (residues 1-505; isoform 2) inserted in pCMV6-XL4 plasmid.

Mutant proteins were generated using site directed mutagenesis of the aforementioned plasmids using primers listed in Supplementary Table 2. Peptides: LXXLL-containing motifs from TRAP220 (residues 638–656; NTKNHPMLMNLLKDNPAQD) and NCoR1 (2256–2278; DPASNLGLEDIIRKALMGSFDDK), amidated at the C-terminus for stability, without a FITC-label or containing a N-terminal FITC label with a six-carbon linker (Ahx), or NCoR1 (2251-2273; GHSFADPASNLGLEDIIRKALMG) containing an N-terminal biotin label and an amidated C-terminus for stability, were synthesized by LifeTein.

**Cell-based transcriptional reporter assays**. HEK293T cells (ATCC; authenti-cated by morphology) were cultured in DMEM medium supplemented with 10% fetal bovine serum (FBS) and 50 units ml$^{−1}$ of penicillin, streptomycin, and glu-tamine. Cells were grown to 90 % confluency and then seeded in 10 cm dishes at 4 million cells per well. Cells were transfected using X-tremegene 9 (Roche) and Opti-MEM (Gibco) with pCMV6 full-length PPARγ expression plasmid (4.5 μg) and 3xPPRE-lucifease reporter pGL2 plasmid (4.5 μg) and incubated for 18 h; plasmids were obtained from P. Griffin (Scripps) as used in previous studies.[3,6,15,32] Cells were transferred to white 384-well plates (Thermo Fisher Scientific) at 10,000 cells per well in 20 μL and incubated for 4 h. Ligand (5 μM) or vehicle control was added (20 μL), cells incubated for 18 hr and harvested for luciferase activity quantified using Britelite Plus (Perkin Elmer; 20 μL) or cell viability was tested using Celltiter-glo (Promega; 20 μL) on a Synergy Neo multimode plate reader (BioTek). Data were analyzed using GraphPad Prism (luciferase activity vs. ligand concentration) and fit to a sigmoidal dose response curve. For western blot analysis of protein levels, HEK293T cells were transfected as described above. Following transfection, 250,000 cells were transferred to 6-well plates (Corning), incubated for 4 h, treated with 5 μM ligand, and incubated overnight. Transfected cells were then lysed in TNT buffer (150 mM NaCl, 0.1 M Tris, 0.1% Tween 20, pH 7.5) and incubated for 1 h at 4 °C. Protein concentration was determined by BCA assay (Thermo Fisher), and 20 μg of protein was loaded onto 4–15% gradient gels (Bio-Rad) and wet transferred onto PVDF. The membrane was blocked for 1 h at room temperature with Odyssey blocking buffer (Li-COR). After blocking, the mem-brane was incubated overnight at 4 °C with primary antibodies diluted in Odyssey blocking buffer: 1:1000 rabbit anti-PPARγ (Cell signaling technology; catalog #2443 S, lot # 6), 1:1000 mouse anti-actin (EMD Millipore; catalog #MAB1501, lot # 2895655). The following day the blot was washed with PBST and treated with secondary antibodies (Li-COR; donkey anti-mouse-IgG IRDye 680LT, catalog #925-68022, lot # C71201-15; goat anti-rabbit-IgG IR800CW, catalog #926-32211, lot # C80546-08) at 1:2000 dilution in Odyssey blocking buffer for 1 hr at RT. The blot was then washed with PBST and visualized via multiplexed detection using the Odyssey 9120 infrared imaging system (Li-COR).

**3T3-L1 cell gene expression analysis**. 3T3-L1 cells (gift from Anutosh Chakra-borty; authenticated by morphology) were cultured in DMEM medium supple-mented with 10% FBS and 50 units ml$^{−1}$ of penicillin, streptomycin, and glutamine. Cells were grown to 90% confluency and then seeded in 12-well dishes at 50,000 cells per well and incubated overnight at 37 °C, 5% CO$_2$. The following day, cells were treated with media supplemented with 0.5 mM 3-iso-butyl-1-methylxanthine, 1 μM dexamethasone, and 877 nM insulin. Following 2-days of incubation, cells were treated with 10 μM compound in media supplemented with 877 nM insulin for 24 h. RNA was extracted using quick-RNA MiniPrep Kit (Zymo) and used to generate complementary DNA using qScript cDNA synthesis kit (Quantabio). Expression levels of PPARγ target genes were measured using Applied Biosystems 7500 Real-Time PCR system. Relative gene expression of *aP2/FABP4* and *CD36* was calculated using the ddCt method after normalization to *TBP* using primers listed in Supplementary Table 3 and plotted using GraphPad Prism.

**Fluorescence polarization coregulator interaction assay**. The assay was performed in black 384-well plates (Greiner) in assay buffer (20 mM potassium phosphate, 50 mM potassium chloride, 1 mM TCEP, 0.01% Tween 20, pH 7.4). His-PPARγ LBD was pre-incubated with or without a 2X molar excess of covalent ligand overnight at 4 °C, diluted by serial dilution, and plated. Noncovalent compounds were incubated with a constant concentration of 90 μM, equivalent to the maximum protein concentration, to ensure full occupancy. FITC-labeled NCoR1 and TRAP220 peptides were plated at a final concentration of 100 nM. Plates were incubated for 2 h at 4 °C and measured on a Synergy Neo multimode plate reader (BioTek) exciting at 495 nm and reading at 528 nm wavelengths. Data were plotted using GraphPad Prism and fit to one-site binding equation. We observed no significant changes in coregulator affinity when using native or delipidated protein, indicating any bacterial lipids retained during protein purification are bound substoichiometrically. For the assay using full-length protein, FITC-labeled NCoR1 peptide was plated to a final concentration of 50 nM in wells to which was added delipidated full-length PPARγ loaded with a stoichiometric amount of ligand (protein concentration ranged from 50 μM to 24 nM by a 12 point 2-fold dilution) in a buffer containing 25 mM MOPS (pH 7.4), 25 mM potassium chloride, 1 mM EDTA, 0.01% fatty acid-free bovine serum albumin (BSA) (Millipore), 0.01% Tween, and 5 mM TCEP. Plates were incubated in the dark for 2 h at room temperature and measured on a Synergy H1 microplate reader (BioTek) exciting at 495 nm and reading at 528 nm wavelengths.

**Time-resolved fluorescence resonance energy transfer assay**. A 2X stock of His-PPARγ LBD, LanthaScreen® Elite Tb-anti-HIS Antibody (ThermoFisher; catalog #PV5863, lot # 1730002 A), and FITC-labeled NCoR1 peptide was prepared in assay buffer (20 mM potassium phosphate, 50 mM potassium chloride, 1 mM TCEP, 0.01% Tween 20, pH 7.4). Ligands were prepared as a 2X ligand stock, initially prepared via serial dilution in DMSO prior to addition to buffer. Equal volumes of the protein mixture and ligand were added to black 384 well plates (Greiner) for final concentrations of 4 nM protein, 400 nM peptide, and 1 nM antibody. Final concentration of DMSO (vehicle) was constant in all wells at 0.5%. Plates were incubated for 2 h at 4 °C and measured on a Synergy Neo multimode plate reader (BioTek). The Tb donor was excited at 340 nm, its emission was measured at 495 nm, and the acceptor FITC emission was measured at 520 nm. Data were plotted using GraphPad Prism (TR-FRET ratio 520 nm/495 nm vs. ligand concentration) and fit to sigmoidal dose response equation. For the full-length PPARγ assay, the buffer contained 25 mM MOPS (pH 7.4), 25 mM potassium chloride, 1 mM EDTA, 0.01% fatty acid-free BSA (Millipore), 0.01% Tween, and 5 mM TCEP. Delipidated His-full-length PPARγ/full-length RXRα were plated at 31 nM with 0.9 nM LanthaScreen® Elite Tb-anti-HIS Antibody, 200 nM biotin-labeled NCoR1 peptide, 400 nM streptavidin-d2 (Cisbio, 610SADLB), and 12-point serial dilutions of T0070907 from 50 μM to 1 pM to a final volume of 16 μL. Plates were incubated for 4 h at room temperature and measured on Synergy H1 microplate reader (BioTek) at 620 nm for terbium and 665 nm for d2. After reading, a 1.25x molar excess of dsDNA of the Sult2A1 PPRE (5′-GTA AAA TAG GTG AAA GGT AA-3′; and its reverse complement) was added to each well, and the plate was incubated for 3 h prior to a subsequent reading. Data were plotted using GraphPad Prism (TR-FRET ratio 665 nm/620 nm vs. ligand concentration) and fit to a sigmoidal dose response equation.

**Crystallography data collection and structure determination**. PPARγ LBD protein was concentrated to 10 mg ml$^{-1}$ and buffer exchange into phosphate buffer (20 mM potassium chloride, 50 mM potassium phosphate, 5 mM TCEP, pH 8.0). Crystals of T0070907-bound PPARγ were obtained by soaking the ligand into preformed apo-protein crystals since our cocrystallization attempts failed. Apo-PPARγ crystals were obtained after 3–5 days at 22 °C by sitting-drop vapor diffusion against 50 μl of well solution using 96-well format crystallization plates. The crystallization drops contain 1 μl of protein sample mixed with 1 μl of reservoir solution containing 0.1 M MOPS, 0.8 M sodium citrate at pH 6.5. T0070907 was soaked into the PPARγ apo-crystals drop by adding 1 μl of compound at a concentration of 1 mM suspended in the same reservoir solution for 3 weeks. Crystals were cryoprotected by immersion in mother liquor containing 12% glycerol and flash-cooled in liquid nitrogen before data collection. Data collection was carried out at Beamlines 5.0.1 of BCSB at the Advanced Light Source (Lawrence Berkeley National Laboratory). Data were processed, integrated, and scaled with the programs Mosflm and Scala in CCP4[47,48]. The structure was solved by molecular replacement using the program Phaser[49] implemented in the PHENIX package[50] and used previously published PPARγ LBD structure (PDB code: 1PRG)[51] as the search model. The structure was refined using PHENIX with several cycles of interactive model rebuilding in COOT[52].

**Molecular dynamics simulations**. A crystal structure of GW9662-bound PPARγ (PDB code 3B0R) along with our crystal structure T0070907 bound to PPARγ (PDB code 6C1I) were used to build initial structures in all simulations in this study. Two models were generated using 3B0R crystal structure. In the first model, chain A of 3B0R was used and GW9662 was transformed to T0070907 by converting phenyl ring of GW9662 to the pyridine ring of T0070907. The second 3B0R generated model was built using chain B conformation. In addition, chain B of the

T0070907 crystal structure was used for a third build. The crystalized water molecules were kept in the models in which chain B conformations were used. The Modeller[53] extension within UCSF Chimera[54] was used to fill in the missing part of the protein in PDB files. The resulting structures were submitted to H + + server[55] to determine the protonation states of titratable residues at pH 7.4. AMBER names were assigned to different protonation states of histidine using pdb4amber provided in AmberTools 14[3]. In order to parametrize T0070907 and GW9662, the C285 with covalently attached ligand was protonated and methyl caps were added, saved as a separate PDB file using Chimera, and submitted to the R.E.D server[56] to calculate RESP[57] charges. AMBER cysteine residue values were used for the RESP charges for the cysteine backbone. The output mol2 file was used to generate the ac and prepin files following a method in the tutorial (http://ambermd.org/tutorials/basic/tutorial5/). Parmchk2 was used to create two force modification files from the prepin file, one that used AMBER ff14SB[58] parameter database values and another that used general Amber force field[59] (GAFF2) values, then Tleap was used to generate topology and coordinate files. The ff14SB force field was used to describe the protein. The resulting structure was solvated in a truncated octahedral box of TIP3P water molecules with the 10 Å spacing between the protein and the boundary, neutralized with Na+ and K+ and Cl− ions were added to 50 mM. The system was minimized and equilibrated in nine steps at 310 K with nonbonded cutoff of 8 Å. In the first step the heavy protein atoms were restrained by a spring constant of 5 kcal mol$^{-1}$ Å$^{-2}$ for 2000 steps, followed by 15 ps simulation under NVT conditions with shake, then two rounds of 2000 cycles of steepest descent minimization with 2 and 0.1 kcal mol$^{-1}$ Å$^{-2}$ restraints were performed. After one round without restraints, three rounds of simulations with shake were conducted for 5 ps, 10 ps, and 10 ps under NPT conditions and restraints of 1, 0.5, and 0.5 kcal mol$^{-1}$ Å$^{-2}$ on heavy atoms. Finally, an unrestrained NPT simulation was performed for 200 ps. Production runs were carried out with hydrogen mass repartitioned[60] parameter files to enable 4 fs time steps. Constant pressure replicate production runs were carried out with independent randomized starting velocities. Pressure was controlled with a Monte Carlo barostat and a pressure relaxation time (taup) of 2 ps. Temperature was kept constant at 310 K with Langevin dynamics utilizing a collision frequency (gamma_ln) of 3 ps$^{-1}$. The particle mesh ewald method was used to calculate non-bonded atom interactions with a cutoff (cut) of 8.0 Å. SHAKE[61] was used to allow longer time steps in addition to hydrogen mass repartitioning. Production simulations were run in triplicate or quadruplicate for the following durations—T0070907 (modeled conformation; 3 total): 24.3, 26.4, and 13.4 μs; T0070907 (crystallized conformation; 4 total): all were 4 μs; GW9662 (crystallized conformation; 4 total): all were 4 μs. Analysis of trajectories was performed using cpptraj[62]. Hydrogen bond analysis was performed using dist = 3.5 Å and angle = 100°[63].

**Circular dichroism spectroscopy**. Protein samples pre-incubated with or without a 2X molar excess of covalent ligand overnight at 4 °C were diluted to 10 μM in CD buffer (10 mM potassium phosphate, 50 mM potassium fluoride, pH 7.4) and measured on a JASCO J-815 CD spectrometer by scanning from 190 nm to 300 nm at 20 °C or by increasing the temperature from 20 to 80 °C (at 1 °C min$^{-1}$) while monitoring the CD signal at 222 nm. Protein unfolding/melting temperature ($T_m$) was determined by fitting the data to a thermal unfolding equation[64] in GraphPad Prism.

**NMR spectroscopy**. NMR data on $^{15}$N-labeled PPARγ LBD (wild-type or mutants, generally at 200 μM), with or without pre-incubation with a 2X molar excess of covalent ligand overnight at 4 °C, were acquired at 298 K (unless otherwise indicated) in NMR buffer (20 mM potassium phosphate, 50 mM potassium chloride, 1 mM TCEP, pH 7.4, 10% D$_2$O) on a Bruker 700 MHz NMR instrument equipped with a QCI -P cryoprobe or, in the case of ZZ-exchange experiments, on a Bruker 800 MHz NMR instrument equipped with a TCI cryoprobe. For peptide titrations, peptides were dissolved in NMR buffer and added at the molar equivalents indicated in the figures. Temperature-dependent data were referenced indirectly using DSS (4,4-dimethyl-4-silapentane-1-sulfonic acid). ZZ-exchange experiments were acquired at 298 K or 310 K on Bruker 700 or 800 MHz NMR instrument equipped with a QCI or TCI cryoprobe, respectively, using exchange mixing times ranging from 0–2 s. The 3D $^{15}$N-NOESY-HSQC experiment was performed using a mixing time of 100 ms. The 2D CLEANEX-PM experiment[28] was performed using a mixing time of 100 ms. Data were processed and analyzed using Topspin (version 3; Bruker Biospin) and NMRViewJ (version 9.2; OneMoon Scientific, Inc.)[65], respectively. NMR chemical shift assignments previously described for ligand-bound PPARγ[6,15,32] (BMRB accession codes 17975, 17976, 17977) were assigned to the spectra for well-resolved residues with consistent NMR peak positions the presence of different ligands using the minimum chemical shift perturbation procedure[66]. ZZ-exchange data were fit to an exchange model for slow two-state interconversion[33,67] using a protocol and a MATLAB script provided by Gustafson, et al.[68] along with MATLAB software (version R2018a). LineShapeKin[35], implemented in MATLAB scripts, was used to simulate 1D NMR line shapes using MATLAB software (version R2018a) from the peptide titration experiments for G399 ($^{15}$N planes vs. extracted $^{15}$N planes from 2D NMR data) using 2-state (U), 3-state (U_RL), and 4-state (U-R-RL) models. All simulations were performed using the 4-state model MATLAB script. For the 2-state and 3-state simulations, parts of the 4-state model were turned off as described in the

LineShapeKin manual. The 4-state simulations were performed considering the exchange rates and molar fractions of the two slowly exchanging PPARγ populations from the ZZ-exchange analysis.

For $^{19}$F NMR, PPARγ LBD K474C mutant protein was used to allow covalent attachment of 3-bromo-1,1,1-trifluoroacetone (BTFA) helix 12 via K474C. Mass spectrometry verified that GW9662 and T0070907 (2X molar excess) do not covalently attach to K474C (using a K474C/C285S double mutant protein that is in capable of covalent attachment to C285); using wild-type protein confirmed covalently attachment to C285. Samples were first incubated with 2X GW9662 or T0070907, then incubated with 2X BTFA, followed by buffer exchange into $^{19}$F NMR buffer (25 mM MOPS, 25 mM potassium chloride, 1 mM EDTA, pH 7.4, 10% D$_2$O). 1D $^{19}$F NMR data of 150 μM BTFA-labeled PPARγ LBD bound to GW9662 or T0070907 were acquired at 298 K on a Bruker 700 MHz NMR instrument equipped with QCI-F cryoprobe. Chemical shifts were calibrated using an internal separated potassium fluoride reference in $^{19}$F NMR buffer without TCEP contained in a coaxial tube inserted into the NMR sample tube set to be −119.522 ppm, which is the chemical shift of the signal with respect to the $^{19}$F basic transmitter frequency for instrument. 1D $^{19}$F spectra were acquired utilizing the zgfhigqn.2 pulse program provided in Topspin 3.5 (Bruker Biospin). Data were processed using Bruker Topspin (version 3; Bruker Biospin) and deconvoluted with decon1d[69] (version 2; https://github.com/hughests/decon1d).

**Code availability**. Scripts or non-commercially available programs used for the analysis in this study are publicly available, including MATLAB scripts used for the ZZ exchange NMR lineshape analyses [https://osf.io/4fmkd/], MATLAB from the LinShapeKin package [http://lineshapekin.net], and decon1d [https://github.com/hughests/decon1d].

## Data availability

The crystal structure of T0070907-bound PPARγ LBD is available in the Protein Data Bank (PDB accession code 6C1I). NMR chemical shift assignments used in our analysis were obtained from the Biological Magnetic Resonance Data Bank (BMRB accession codes 17975, 17976, 17977). Any other datasets generated during and/or analyzed during the current study are available from the corresponding author on reasonable request.

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

## Acknowledgements

We thank Paola Munoz-Tello for critical reading of the manuscript. This work was supported by National Institutes of Health (NIH) grants R01DK101871 (DJK), F32DK108442 (RB), R01DK105825 (PRG), R00DK103116 (TH), and P20GM103546 (computational resources to TH); American Heart Association (AHA) fellowship award 16POST27780018 (RB); and the William R. Kenan, Jr. Charitable Trust (TSRI High School Student Summer Internship Program). A portion of this work (ZZ exchange NMR) was performed at the National High Magnetic Field Laboratory (NHMFL/MagLab), which is supported by National Science Foundation (NSF) Cooperative Agreement No. DMR-1157490 and the State of Florida. ¹⁹F NMR data presented herein were collected at the CUNY ASRC Biomolecular NMR Facility.

## Author contributions

R.B., J.S., J.F., J.B., A.C., I.M.C., and M.D.N performed mutagenesis and/or purified proteins. R.B., I.M.C., and M.D.N. performed biochemical assays. R.B. performed cellular assays and mutagenesis assays. J.S. and J.F. performed crystallography. R.B., J.S., and D.J. K. performed the 2D/3D NMR experiments. I.M.C. and T.S.H. performed the ¹⁹F NMR experiments. Z.H., T.S.H, and D.J.K. performed the molecular dynamics simulations. S.A. M. contributed to data analysis. A.-L.B., P.R.G., and T.M.K. supplied compounds. R.B. and D.K. conceived the experiments and wrote the manuscript with input from all authors.

## Additional information

**Competing interests:** The authors declare no competing interests.

