## [Peer Review File · Nature Communications]

Reviewers' comments:

Reviewer #1 (Remarks to the Author):

The Authors compared the crystal structures of PPAR γ complexed with two covalent ligands only differing by a methin (CH) to nitrogen substitution and found that, despite no major structural differences, the two ligands show completely different agonist properties (one, GW9662, is a neutral antagonist, the other, T0070907, an inverse agonist).

They found that the inverse agonist T0070907 (with the nitrogen substitution) realizes a water-mediated H-bond network that uniquely links R288 of PPAR γ to the ligand pyridyl group.

The Authors confirmed the stability of this network by MD simulations. Then, they used mutagenesis, activity assays and fluorescence polarization assays, with and without co-regulators, to demonstrate that the unique H-bond network realized by T0070907 is responsible for the inverse agonism of the ligand.

Moreover, the Authors discovered by NMR spectroscopy that PPAR γ exchanges between two long-lived conformations when bound to T0070907, but not GW9662.

Focusing on the ^{15}N -PPAR γ peaks of the residue G399, located near the coregulator hydrophobic cleft, they found that, in the presence of T0070907, one of the two conformations of PPAR γ prepopulates a corepressor-bound state, priming PPAR γ for high affinity corepressor binding.

I found the manuscript very well written and the experiments very well designed and performed.

The work is convincing and the conclusions original.

The authors elegantly demonstrated their hypothesis. Particularly, the NMR studies, in the presence of increasing amounts of coregulators, proved to be very effective and powerful in understanding the ligand mechanism of action and open new routes to design and characterize PPAR γ inverse agonists.

Minor points to be addressed:

Line 33: The authors affirm that "partial or graded agonists do not hydrogen bond to Y473, but mildly stabilize helix 12 via interactions with other regions of the ligand-binding pocket, resulting less pronounced changes in coregulatory affinity and transcriptional activation".

Actually, there are also ligands, not cited by the Authors, that bind to Y473 but show partial agonism character because of unfavorable interactions with residues of helix 3 (Q286) or interactions with different regions of the LBD (H3, H11 and loop before H12) (Pochetti G. et al., JBC (2007) 282: 17314-17324 and Montanari R. et al., JMC (2008) 51: 7768: 7776).

For this reason I would modify the previous sentence in : "partial or graded agonists, generally, do not hydrogen bond to Y473..."

Line 145: In Results the Authors affirm to have performed the MD simulations ranging from 4-26 microseconds in length, but in Methods regarding MD I didn't find a sentence about this length. Could you confirm that the MD simulation lasted 4-26 microseconds?

Lines 209 and 211: The Authors refer to Figures 3F and 3G, but in Figure 3 they are not present. Maybe should be in the text Figure 3D and 3E?

Line 317: D313 on helix 5, and not D311 !

Line 384: maybe "chemical shit" is to be updated...

Line 414: "deconvoluted" and not "deconvoluated".

Dr. Giorgio Pochetti
Istituto di Cristallografia
Area della Ricerca Roma1 - CNR
Italy

Reviewer #2 (Remarks to the Author):

Brust et al. present a broad analysis of the interactions between a novel PPAR γ ligand (T0070907) and the receptor using a broad range of techniques that include cell-based assays, in vitro binding assay, MD simulations, CD & NMR spectroscopy and crystallography. This molecule, T0070907, is a derivative of another PPAR γ ligand, GW9662. The modification from the precursor to T0070907 is subtle – a methine to nitrogen substitution in T0070907. The authors also compare the mode of binding of these two PPAR γ ligands. The authors conclude that unlike GW9662, T0070907 functions as an ‘inverse agonist’, a property that arises principally from a water-mediated H-bond network that links R288 of PPAR γ to the T0070907 pyridyl group. Furthermore, they report that T0070907-bound PPAR γ exists in two conformations, one similar to GW9662-bound PPAR γ and a second that is unique. They conclude that this study has identified a ‘novel structural mechanism.’

Specific concerns:

1. Figure 1 B. The transactivation data does not match any published observations, specifically, the transcriptional response of PPAR γ to the vehicle/solvent DMSO. Several previous studies show little or no observable transactivation in the absence of exogenous agonist (JBC (2000) 275(3) 1883-7; Cell (2004) 116, 417-29; Scientific rep. (2015), 5, 8256). In fact, the study on which much of this manuscript is based also does not show such PPAR γ activity (Fig 2A in JBC (2002) 277(22), 19649 –19657).

2. The authors refer to the molecule T0070907 as an inverse agonist. However, in the original study on this molecule, (JBC (2002) 277(22), 19649 –19657) T0070907 is shown to be an antagonist in cell-based transactivation studies. It is only referred to being an ‘inverse agonist’ of the GST-PPAR ligand binding domain (LBD) in vitro. Given other studies that show that E. coli preparations of PPAR LBD co-purify with E coli lipids within their ligand binding pockets (Mol Cell. (2006) 21(1):1-2), the ‘inverse agonist’ label for T0070907 may be unfounded.

3. There are similar concerns (as with #2 above) with the interpretation of the TRAP220 & NCoR binding data on Fig 3B & 3C.

4. Although the GW9662 ligand is defined as an antagonist, the structure of the GW9662-PPAR γ LBD complex (3B0R) is of PPAR γ in the agonist conformation (helix 12 conformation), suggesting the opposite. This is important because this study derives much of its conclusions in comparison to the GW9662-PPAR γ LBD complex (3B0R) structure.

4. The manuscript is missing Figures 3F and 3G.

5. The NMR study shows the chemical shifts in some distinguishable resonances. However, a significant portion of the peaks are unresolved. It is likely that many of the shifts are conjugated with other movements that are not observed in this study. Ironically, there is no R288, the residue about which this entire study hinges, is unobserved.

Reviewer #3 (Remarks to the Author):

This manuscript addresses structural mechanism for the difference in effect between the highly similar compounds, GW9662 and T0070907, on peroxisome proliferator-activated receptor gamma (PPAR γ) activity. Despite simple methane to nitrogen substitution, GW9662 is a neutral antagonist of PPAR γ , while T0070907 is an inverse agonist. Using crystallography, molecular dynamics (MD) simulations and mutagenesis, they showed that water-mediated hydrogen bond network linking the T0070907 pyridyl group to Arg288 is essential for the inverse agonism. They also performed NMR experiments, and showed that PPAR γ adapts two long-lived conformations when bound to

T0070907 but not GW9662, one of which resembles a GW9662-bound state and the other is a unique state that is similar to the corepressor-bound state.

This manuscript is well-written and provided insights into the mechanism for inverse agonism of PPAR γ . However, I have some concerns as follows.

1. The relationship between the water-mediated hydrogen bond network and the prepopulated corepressor-bound conformation in the T0070907-bound state is not clear. I wonder whether the hydrogen bond network is necessary for adapting the corepressor-bound conformation. This point should be addressed, for example, by NMR measurements of a T0070907-bound Arg288 mutant.

2. If bound water is actually located as shown in Figures 2B and 2C, a NOE signal will be observed between Arg288 H ϵ and water when PPAR γ is bound to T0070907 but not GW9662. The NOE signal will be a strong evidence for the existence of the water-mediated hydrogen bond network.

3. In Figure 4C, why are peaks in the T0070907-bound state at 283K single and weak? Since the exchange rate is slower at a lower temperature, the peaks will be also doubled in 283K in the case of 2-site exchange.

4. Regarding the titration experiments of T0070907-bound PPAR γ with NCoR and TRAP220 peptides, the authors simply explained that the two conformations independently had bound to the peptides. But the state A and B will exchange with each other as shown by the ZZ-exchange experiment. Therefore, during the titrations, a part of state A and B would convert to state B and A, respectively, and then bind to NCoR or TRAP220 peptides. Could not such pathways be observed in the titration experiments? A figure showing the chemical reaction pathway that explains the peak behavior in the titration experiments will be useful to understand the result.

We thank the Reviewers for their time and constructive comments regarding our manuscript. Our revised manuscript includes new data in response to Reviewers 2 and 3 that together with other revisions recommended by all reviewers address the major points raised by the reviewers.

Our original manuscript contained 5 figures, and in this revised manuscript we split the previous version of Figure 5 into two figures (Figure 5, which also contains some new data requested by Reviewer 3; and Figure 6). We also corrected the radar plot for the R288K mutant in Figure 3E; in the original version, we mistakenly used the wild-type (R288) plot for the R288K mutant. Our original supporting information document contained 4 figures and 1 table; the revised version contains 11 figures and 2 tables.

Below, our point-by-point response is formatted as a blue text indented paragraph with a left vertical line. We also included a version of the revised manuscript with Microsoft Word's Track Changes enabled to enable easy visualization of the changes made to the original manuscript.

Reviewer #1 (Remarks to the Author):

The Authors compared the crystal structures of PPAR γ complexed with two covalent ligands only differing by a methine (CH) to nitrogen substitution and found that, despite no major structural differences, the two ligands show completely different agonist properties (one, GW9662, is a neutral antagonist, the other, T0070907, an inverse agonist). They found that the inverse agonist T0070907 (with the nitrogen substitution) realizes a water-mediated H-bond network that uniquely links R288 of PPAR γ to the ligand pyridyl group. The Authors confirmed the stability of this network by MD simulations. Then, they used mutagenesis, activity assays and fluorescence polarization assays, with and without coregulators, to demonstrate that the unique H-bond network realized by T0070907 is responsible for the inverse agonism of the ligand. Moreover, the Authors discovered by NMR spectroscopy that PPAR γ exchanges between two long-lived conformations when bound to T0070907, but not GW9662. Focusing on the ¹⁵N-PPAR γ peaks of the residue G399, located near the coregulator hydrophobic cleft, they found that, in the presence of T0070907, one of the two conformations of PPAR γ prepopulates a corepressor-bound state, priming PPAR γ for high affinity corepressor binding. I found the manuscript very well written and the experiments very well designed and performed. The work is convincing and the conclusions original. The authors elegantly demonstrated their hypothesis. Particularly, the NMR studies, in the presence of increasing amounts of coregulators, proved to be very effective and powerful in understanding the ligand mechanism of action and open new routes to design and characterize PPAR γ inverse agonists.

Authors' response: We thank the reviewer for their comments and for pointing out the minor points listed below, which we have addressed in this revised manuscript.

Minor points to be addressed:

Line 33: The authors affirm that “partial or graded agonists do not hydrogen bond to Y473, but mildly stabilize helix 12 via interactions with other regions of the ligand-binding pocket, resulting less pronounced changes in coregulatory affinity and transcriptional activation”. Actually, there are also ligands, not cited by the Authors, that bind to Y473 but show partial agonism character because of unfavorable interactions with residues of helix 3 (Q286) or interactions with different regions of the LBD (H3, H11 and loop before H12) (Pochetti G. et al., JBC (2007) 282:17314-17324 and Montanari R. et al., JMC (2008) 51:7768:7776). For this reason I would modify the previous sentence in : “partial or graded agonists, generally, do not hydrogen bond to Y473...”

Authors' response: These studies are indeed interesting and adds to the known structural mechanisms for directing PPAR γ partial agonism. We modified our introduction to include a discussion of this mechanism and cited the above references.

Line 145: In Results the Authors affirm to have performed the MD simulations ranging from 4-26 microseconds in length, but in Methods regarding MD I didn't find a sentence about this length. Could you confirm that the MD simulation lasted 4-26 microseconds?

Authors' response: We have updated our Methods section to include this information as follows:

“Production simulations were run in triplicate or quadruplicate for the following durations— T0070907 (modeled conformation; 3 total): 24.3, 26.4, and 13.4 μ s; T0070907 (crystallized conformation; 4 total): all were 4 μ s; GW9662 (crystallized conformation; 4 total): all were 4 μ s.”

Lines 209 and 211: The Authors refer to Figures 3F and 3G, but in Figure 3 they are not present. Maybe should be in the text Figure 3D and 3E?

Authors' response: We corrected this typographical error, which should have referred to Figure 3D and 3E.

Line 317: D313 on helix 5, and not D311 !

Authors' response: We corrected this to read D313.

Line 384: maybe “chemical shit” is to be updated...

Authors' response: We appreciate that the reviewer caught this typographical error and corrected it to “chemical shift”.

Line 414: “deconvoluted” and not “deconvoluted”.

Authors' response: We corrected this typographical error.

Dr. Giorgio Pochetti
Istituto di Cristallografia
Area della Ricerca Roma1 - CNR
Italy

Reviewer #2 (Remarks to the Author):

Brust et al. present a broad analysis of the interactions between a novel PPAR γ ligand (T0070907) and the receptor using a broad range of techniques that include cell-based assays, in vitro binding assay, MD simulations, CD & NMR spectroscopy and crystallography. This molecule, T0070907, is a derivative of another PPAR γ ligand, GW9662. The modification from the precursor to T0070907 is subtle – a methine to nitrogen substitution in T0070907. The authors also compare the mode of binding of these two PPAR γ ligands. The authors conclude that unlike GW9662, T0070907 functions as an ‘inverse agonist’, a property that arises principally from a water-mediated H-bond network that links R288 of PPAR γ to the T0070907 pyridyl group. Furthermore, they report that T0070907-bound PPAR γ exists in two

conformations, one similar to GW9662-bound PPAR γ and a second that is unique. They conclude that this study has identified a ‘novel structural mechanism.’

Authors’ response: We appreciate the reviewer’s comments below, which have given us the opportunity to better clarify these points for the general readership in our revised manuscript.

Specific concerns:

1. Figure 1 B. The transactivation data does not match any published observations, specifically, the transcriptional response of PPAR γ to the vehicle/solvent DMSO. Several previous studies show little or no observable transactivation in the absence of exogenous agonist (JBC (2000) 275(3) 1883-7; Cell (2004) 116, 417-29; Scientific rep. (2015), 5, 8256). In fact, the study on which much of this manuscript is based also does not show such PPAR γ activity (Fig 2A in JBC (2002) 277(22), 19649 –19657).

Authors’ response: In our revised manuscript, we include a new supplementary figure (Figure S2; see the image to the right), which clearly shows that full-length PPAR γ shows a transcriptional response (i.e., basal activity) in the absence of an exogenously added ligand (i.e., the vehicle/solvent DMSO control condition) relative to transfection of an empty control plasmid. Also, in Figure 1C we now show new qPCR gene expression data from 3T3-L1 preadipocyte cells treated with the same ligands used in the cell-based transactivation and biochemical profiling assays. The ligands show the same profiles in the expression of two known PPAR γ target genes, FABP4/aP2 and CD36; and in both cases T0070907 represses the expression of these PPAR γ target genes relative to DMSO control treated cells. These observations are consistent with our previous work published in *Nature Communications*, which showed that the direct antagonist ligands (SR2595 and SR10221) that we also studied in this report decrease PPAR γ transactivation (transcriptional repression) relative to vehicle (DMSO) control, which we similarly show in Figure 1B of this current manuscript. The only way transcriptional repression can be observed in these assays (reporter transactivation assay and the qPCR assay) is because PPAR γ has significant observable basal transactivation in the absence of exogenous ligand. One reason that PPAR γ has basal activity in the vehicle (DMSO) control condition is cells contain endogenous PPAR γ ligands (lipids and fatty acids), which act as PPAR γ agonists by enhancing coactivator binding and decreasing corepressor binding—as we show in Figure 1D for nonanoic acid.

Second, our data do match published observations and the general knowledge in the field that PPAR γ has transcriptional activity in cells without exogenously added ligand. There are numerous publications showing this for other PPARs and other nuclear receptors. We found several publications that similarly observed an increase in basal activity upon cellular transfection of full-length PPAR γ :

1. Figure 5 in Welters HJ, McBain SC, Tadayyon M, Scarpello JH, Smith SA, Morgan NG. Expression and functional activity of PPAR γ in pancreatic beta cells. *Br J Pharmacol.* 2004;142(7):1162-70.
2. Figure 5B in Hou Y, Moreau F, Chadee K. PPAR γ is an E3 ligase that induces the degradation of NF κ B/p65. *Nat Commun.* 2012;3:1300.
3. Figure 1A in Wang C, Pattabiraman N, Zhou JN, Fu M, Sakamaki T, Albanese C, Li Z, Wu K, Hulit J, Neumeister P, Novikoff PM, Brownlee M, Scherer PE, Jones JG, Whitney KD, Donehower LA, Harris EL, Rohan T, Johns DC, Pestell RG. Cyclin D1 repression of

peroxisome proliferator-activated receptor gamma expression and transactivation. *Mol Cell Biol.* 2003;23(17):6159-73.

4. Figure 3 in Al-Rasheed NM, Chana RS, Baines RJ, Willars GB, Brunskill NJ. Ligand-independent activation of peroxisome proliferator-activated receptor-gamma by insulin and C-peptide in kidney proximal tubular cells: dependent on phosphatidylinositol 3-kinase activity. *J Biol Chem.* 2004;279(48):49747-54.

and/or cellular treatment of a repressive ligand:

1. Figure 5A in Huang C, Zhang Y, Gong Z, Sheng X, Li Z, Zhang W, Qin Y. Berberine inhibits 3T3-L1 adipocyte differentiation through the PPARgamma pathway. *Biochem Biophys Res Commun.* 2006;348(2):571-8.

We carefully read the four studies cited by the reviewer. One article cited by the reviewer does not focus on PPAR γ or a nuclear receptor in general (*Scientific rep.* 2015, 5, 8256); it is titled “Efficient generation of gene-modified pigs via injection of zygote with Cas9/sgRNA”. For another citation provided by the reviewer, we could not find an article under the citation provided (*JBC* 2000, 275(3), 1883-7).

In the *Cell* (2004) 116, 417-29 paper, we could not find any cellular transactivation data for PPAR γ , but data are shown for a related receptor, PPAR α , in Figure 4D. Consistent with our findings for PPAR γ in this manuscript, PPAR α showed basal activity of ~200 RLU units in the *Cell* paper.

In the original report of T0070907 (*JBC* 2002, 277, 22:19649 – 19657), a Gal4-PPAR γ LBD chimeric fusion protein was used to in Figure 2A to show that T0070907 inhibits rosiglitazone’s ability to activate the Gal4-PPAR γ LBD chimeric fusion. A close inspection of the data in Figure 2A in this *JBC* paper shows that the bar graph for T0070907 treated cells is lower than basal expression, which is consistent with transcriptional repression. *To the right, we have cropped and increased the size of Figure 2A from this JBC paper; the red and blue arrows indicate conditions where cells were treated with T0070907 alone (red arrow) or T0070907 cotreated with rosiglitazone (blue arrow). Note that the window for repression is masked because of the DMSO control-normalized Y-axis units; a 10-fold decrease in luciferase activity would appear smaller (RLU=0.1) than a 10-fold increase (RLU=10) unless the Y-axis units were log-transformed then they would appear equivalent.* Consistent with our data, in both cases the luciferase activity for T0070907 treated cells is lower than vehicle control (*1st white bar in the JBC Figure 2A excerpt to the right*). It is important to note that in this *JBC* study, T0070907 was cotreated with rosiglitazone because the main point of the paper is T0070907 inhibits binding of other synthetic ligands—this provided the initial “antagonist” label. However, the ability of T0070907 to inhibit the binding and activity of another ligand is not a pharmacological description of T0070907; the use of the term “antagonist” was not descriptive of T0070907’s transcriptional properties on its own.

2. The authors refer to the molecule T0070907 as an inverse agonist. However, in the original study on this molecule, (*JBC* (2002) 277(22), 19649 –19657) T0070907 is shown to be an antagonist in cell-based transactivation studies. It is only referred to being an ‘inverse agonist’ of the GST-PPAR ligand binding

domain (LBD) in vitro. Given other studies that show that *E. coli* preparations of PPAR LBD co-purify with *E. coli* lipids within their ligand binding pockets (Mol Cell. (2006) 21(1):1-2), the ‘inverse agonist’ label for T0070907 may be unfounded. 3. There are similar concerns (as with #2 above) with the interpretation of the TRAP220 & NCoR binding data on Fig 3B & 3C.

Authors’ response: As mentioned above, GW9662 and T0070907 were referred to as antagonists in their original studies because they bind covalently and block binding of other synthetic ligands such as rosiglitazone. However, the ability of a ligand such as GW9662 or T0070907 to block other ligands from binding to the orthosteric pocket is not a pharmacological description of their transcriptional activities. Our manuscript explores the pharmacological properties of GW9662 and T0070907 in head-to-head cellular (transactivation reporter and QPCR gene expression) and coregulator interaction assays by comparing T0070907 to GW9662 and other activating and repressive PPAR γ ligands, which clearly shows T0070907 is an inverse agonist by definition: an inverse agonist enhances corepressor binding and decreases coactivator binding. In contrast, an antagonist by definition is a ligand that blocks the activity of other ligands while lacking the ability to modulate the receptor’s activity on their own. This would mean that the activity of PPAR γ bound to an antagonist should be similar to ligand free/apo-PPAR γ . We added text to the manuscript introduction, results, and discussion sections to better describe the difference between GW9662’s and T0070907’s ability to inhibit ligand binding, which led to their original “antagonist” label; and their ability to influence PPAR γ transcription, target gene expression, and coregulator interaction on their own, which provides their true pharmacological labels. We included a new supporting information table (**Table S1**) that provides pharmacological definitions of nuclear receptor agonists, neutral antagonists, direct antagonists, and inverse agonists.

The reviewer indicated the original *JBC* paper only showed T0070907 to be an inverse agonist within the context of the GST-tagged PPAR γ LBD. However, the *JBC* paper also showed that T0070907 increased interaction of full-length NCoR to full-length PPAR γ /RXR α heterodimer bound to DNA using an GMSA/EMSA assay (Figure 5 in the *JBC* paper). Consistent with these results, we provide new data in our manuscript showing that T0070907 strengthens the affinity (K_d) of our NCoR peptide for full-length PPAR γ using a fluorescence polarization interaction assay (**Figure S3**) and enhances the binding of our NCoR peptide to the PPAR γ LBD and the full-length PPAR γ /RXR α heterodimer bound to a PPAR γ response element (PPRE) DNA sequence (**Figure S4**).

Finally, using our fluorescence polarization coregulator interaction assay we found that delipidation of PPAR γ LBD protein does not significantly affect coregulator affinity (see the figure to the right), likely because any co-purified bacterial lipids are bound substoichiometrically

because of our extensive purification methods where any bound lipids would likely exchange off during column chromatography. This is supported by our coregulator profiling data we showed in **Figure 1D,E**, where addition of nonanoic acid—a human dietary PPAR γ ligand that is also present in bacteria and can be co-purified with PPAR γ (ref. 22 in our manuscript)—causes a significant change in the affinity of coregulator peptides consistent with the definition of an agonist: it enhances coactivator binding and decreases corepressor binding vs. delipidated apo-protein. This observation is consistent with other published studies showing that lipids and fatty

acids are endogenous agonists that function by forming hydrogen bonds with Tyr473 and other residues near helix 12, which stabilizes the AF-2 surface similar to synthetic agonists. If our purified protein was saturated with bacterial lipid, we would not have observed such a dramatic effect upon addition of nonanoic acid in the coregulator profiling assay. Furthermore, our data show that the inverse agonist coregulator recruitment properties of T007 is enhanced, not unfounded as suggested by the reviewer, when compared to PPAR γ bound to lipid (as we showed in Figure 1D) because lipids function as PPAR γ agonists not inverse agonists or neutral antagonists.

4. Although the GW9662 ligand is defined as an antagonist, the structure of the GW9662-PPAR γ LBD complex (3B0R) is of PPAR γ in the agonist conformation (helix 12 conformation), suggesting the opposite. This is important because this study derives much of its conclusions in comparison to the GW9662-PPAR γ LBD complex (3B0R) structure.

Authors' response: As we stated in our manuscript and show in **Figure S8** (previously Figure S2; and we updated this figure to show nearby symmetry related molecules), the “active” or “agonist” conformation of helix 12 is a crystallization artifact. In many of the published ligand-bound PPAR γ crystal structures, helix 12 of chain B docks into the AF-2 surface of chain A, which artificially forces chain A helix 12 into the “agonist” conformation. We discussed this in our manuscript as part of the text that leads up to our NMR studies—and we would argue that it is our comparative NMR studies, not the comparison of the crystalized helix 12 conformations, that is important in deriving the conclusions in our study. In short, as we stated in the manuscript, the crystal structures were critical to identify the pyridyl-water-Arg288 interaction, but our NMR studies were essential to show what happens to the conformation of PPAR γ *in solution* (not in the crystalline state) when bound to T0070907 or GW9662—that T0070907 populates a unique corepressor-like conformation *in solution*, which due to packing interactions in the crystal are not able to be observed in the *crystalline/crystal structure* state.

4. The manuscript is missing Figures 3F and 3G.

Authors' response: We mistakenly called out Figure 3F and 3G; these callouts should have been to Figure 3D and 3E. We thank the reviewer for pointing this out and we corrected this error.

5. The NMR study shows the chemical shifts in some distinguishable resonances. However, a significant portion of the peaks are unresolved. It is likely that many of the shifts are conjugated with other movements that are not observed in this study. Ironically, there is no R288, the residue about which this entire study hinges, is unobserved.

Authors' response: PPAR γ is a relatively large protein to study by NMR, and therefore there is a good amount of NMR chemical shift (peak) overlap in the center of the spectrum, which includes the backbone resonance for R288, making analysis difficult or impossible. In situations like these, it is common to analyze well resolved or “distinguishable” resonances—and to this point, we would also point out that in **Figure S9** (previously Figure S4) we showed that 11 other well-resolved NMR peaks clearly show slow exchange properties on the NMR time scale. Furthermore, in the analysis of ZZ exchange data, the appearance of exchanging crosspeaks typically only occurs for a few resonances when very strict criteria are met: when the two conformations that are exchange have large separation in the ^{15}N dimension. This is best stated in a well-accepted review on NMR experiments to study protein dynamics:

“EXSY may be limited by spectral crowding and/or poor sensitivity because it functions by introducing additional, often weak, signals into the spectrum. Practically though, many EXSY

studies only require a few structural probes to address the questions of interest (as opposed to the tens of structural probes typically required for other NMR-based techniques).

* Reference: Kleckner, I.R., and Foster, M.P. (2011). An introduction to NMR-based approaches for measuring protein dynamics. *Biochim Biophys Acta* 1814, 942-968.

Finally, we note that in an NMR experiment we performed in response to Reviewer 3's critique, we show new NMR data where we assigned the R288 side-chain N ϵ /H ϵ NMR peak (by mutagenesis), determined how it is affected when PPAR γ is bound to T0070907 vs. GW9662, and consistent with our T0070907-bound crystal structure we also observed water interactions with the R288 side-chain N ϵ /H ϵ group using NMR experiments.

Reviewer #3 (Remarks to the Author):

This manuscript addresses structural mechanism for the difference in effect between the highly similar compounds, GW9662 and T0070907, on peroxisome proliferator-activated receptor gamma (PPAR γ) activity. Despite simple methane to nitrogen substitution, GW9662 is a neutral antagonist of PPAR γ , while T0070907 is an inverse agonist. Using crystallography, molecular dynamics (MD) simulations and mutagenesis, they showed that water-mediated hydrogen bond network linking the T0070907 pyridyl group to Arg288 is essential for the inverse agonism. They also performed NMR experiments, and showed that PPAR γ adapts two long-lived conformations when bound to T0070907 but not GW9662, one of which resembles a GW9662-bound state and the other is unique state that is similar to the corepressor-bound state. This manuscript is well-written and provided insights into the mechanism for inverse agonism of PPAR γ . However, I have some concerns as follows.

Authors' response: We thank the reviewer for their comments, suggested experiments, and alternate data interpretations, which we have addressed in this revised manuscript and enhances our work.

1. The relationship between the water-mediated hydrogen bond network and the prepopulated corepressor-bound conformation in the T0070907-bound state is not clear. I wonder whether the hydrogen bond network is necessary for adapting the corepressor-bound conformation. This point should be addressed, for example, by NMR measurements of a T0070907-bound Arg288 mutant.

Authors' response: This was an excellent experiment to suggest. In **Figure 5B and Figure S9C**, we show new 2D NMR data for the R288K and R288A mutants. Interestingly, and consistent with our biochemical and cellular mutagenesis findings, the R288K mutant significantly populates the corepressor-like "unique" conformation, whereas the R288A mutant significantly populates the coactivator-like "mutual" conformation that is shared with GW9662. In these studies, we also found that the GW9662-like/mutual/coactivator-like conformation observed for T0070907-bound PPAR γ is less populated in the R288K mutant. In the discussion, we indicate that this may explain why the R288K mutant shows weakened affinity for the TRAP220 coactivator compared to T0070907-bound PPAR γ . Furthermore, we found that the R288A mutant lowly populates the unique/corepressor-like conformation. As we discuss in the results section, this indicates that:

(1) an extended pyridyl-water hydrogen bond network to residues other than R288, which is observed in the T0070907-bound crystal structure and would be present in wild-type PPAR γ and the R288 mutants, may be involved in lowly populating the unique conformation but it is not sufficient for directing inverse agonism; and

(2) the pyridyl-water interaction with R288 (or a positively charged residue) is necessary for significant population of the unique (corepressor-like) conformation and directing inverse agonism.

2. If bound water is actually located as shown in Figures 2B and 2C, a NOE signal will be observed between Arg288 H ϵ and water when PPAR γ is bound to T0070907 but not GW9662. The NOE signal will be a strong evidence for the existence of the water-mediated hydrogen bond network.

Authors' response: To assign the R288 side-chain N ϵ /H ϵ NMR peak, we compared 2D [^1H , ^{15}N]-HSQC of T0070907-bound wild-type PPAR γ and the R288K mutant. One peak in the Arg side-chain N ϵ /H ϵ spectral region disappeared in the R288K mutant (**Figure S5**). Interestingly, this peak is also missing in GW9662-bound wild-type PPAR γ , indicating that the pyridyl-water-R288 network is involved in stabilizing μs -ms time scale motions (intermediate exchange on the NMR time scale) of the R288 side-chain. To determine if there is a water NOE signal to the R288 side-chain N ϵ /H ϵ group, we performed a 3D ^{15}N -NOESY-HSQC experiment on T0070907-bound PPAR γ (**Figure 2D**). In the R288 side-chain N ϵ /H ϵ NOESY strip, a NOE peak is observed at 4.77 ppm, which is consistent with a water NOE. We also confirmed this result using a Phase-Modulated CLEAN chemical EXchange (CLEANEX-PM) NMR experiment (**Figure S6**), which detects water-protein interactions. Because the R288 side-chain N ϵ /H ϵ NMR peak is not visible when bound to GW9662 due to intermediate exchange, we did not collect the 3D ^{15}N -NOESY-HSQC experiment on GW9662-bound PPAR γ . We provide some additional insight into these data in the discussion section.

3. In Figure 4C, why are peaks in the T0070907-bound state at 283K single and weak? Since the exchange rate is slower at a lower temperature, the peaks will be also doubled in 283K in the case of 2-site exchange.

Authors' response: Not all of the peaks in the T0070907-bound state at 283K are single and weak: for most we see multiple T0070907-bound peaks at 283K. However, for a few peaks we observed that line broadening increases (i.e., peak intensity decreases) as the experimental temperature decreases. This likely corresponds to peaks that show intermediate exchange (μs -ms motions) on the NMR timescale. Interestingly, for residue G321 we see line broadening for GW9662-bound PPAR γ and the mutual/GW9662-like state of T0070907-bound PPAR γ , indicating these states may share conformational and dynamical features; we call this out in the results section of our revised manuscript as a lead-in to the ZZ-exchange NMR analysis. Overall, our NMR observations indicate that each of the two long-lived T0070907-bound conformations, which exchange slowly on the ms-s time scale, can also have μs -ms dynamics or intermediate exchange between two or more conformations on the NMR timescale. We added a schematic energy diagram to describe this conformational ensemble in **Figure 4F**.

4. Regarding the titration experiments of T0070907-bound PPAR γ with NCoR and TRAP220 peptides, the authors simply explained that the two conformations independently had bound to the peptides. But the state A and B will exchange with each other as shown by the ZZ-exchange experiment. Therefore, during the titrations, a part of state A and B would convert to state B and A, respectively, and then bind to NCoR or TRAP220 peptides. Could not such pathways be observed in the titration experiments? A figure showing the chemical reaction pathway that explains the peak behavior in the titration experiments will be useful to understand the result.

Authors' response: We thank the reviewer for this comment and agree that the binding observed in the NMR experiments is more complicated than the peptides binding to the two conformations

independently. The pathways described by the reviewer are difficult to observe because the NMR peaks of the free states (not bound to peptide) overlap with the peptide-bound states, but changes in peak intensities that occur during the titration do support the pathways. As such, we modified our results section and added **Figures S10 and S11** to describe this more likely binding scenario. We added the following text in describing the NCoR titration:

“Because our ZZ-exchange NMR analysis showed that, in the absence of coregulator peptide, states *A* and *B* are in equilibrium on the slow exchange NMR time scale, the initial decrease in the intensity of peak *A* that occurs up to 0.6 equiv NCoR likely corresponds to a conformational redistribution of the peptide unbound states. That is, as NCoR binds to and decreases the population of state *B*, part of the state *A* population redistributes to and exchanges with the state *B* conformation on a time scale of seconds (**Figure S10**). However, visualization of this state *A* → state *B* redistribution is masked, or not readily apparent, because the peaks corresponding to the unbound state *B* and NCoR-bound states overlap.”

And for the TRAP220 titration:

“..., which similar to the NCoR titration is due to the conformational redistribution of states *A* and *B* (**Figure S10**). However, in this case TRAP220 binds to state *A*, which according to the ZZ-exchange analysis has a smaller equilibrium population relative to state *B*, therefore requiring more TRAP220 to fully saturate binding to state *A* in a manner consistent with the TRAP220 equilibrium binding affinity (**Figure S11**).”

Reviewers' comments:

Reviewer #3 (Remarks to the Author):

In the revised manuscript, the authors responded to most of my concerns. However I still have concern about the interpretation of the titration data, which I mentioned before in the comment 4. The explanation of the chemical reaction pathway during the titrations (lines 435-539 and 454-457) is not quite convincing, as the authors did not consider the exchange rates and the molar fraction of each state enough. How about using a computer program such as NMRKIN (Günther, U. L. & Schaffhausen, J. *Biomol. NMR* 22, 201–209 (2002)) or LineShapeKin (Kovrigin, E. L. & Loria, J. P. *Biochemistry* 45, 2636–2647 (2006)) to analyze the titration data in order to determine the most likely chemical reaction pathway and clearly explain the titration data?

A minor point:

(Figure 4F) "T007-bound" will be "T0070907-bound".

Reviewer #4 (Remarks to the Author):

Briefly, the authors found that two structurally similar compounds GW9662 and T0070907, function differently in regulating PPAR γ . While GW9662 blocks the binding of an agonist without recruiting co-repressor to decrease the basal activity of PPAR γ , T0070907 recruits co-repressor to reduce basal activity. To explain the differential activity of GW9662 and T0070907 on basal PPAR γ activity, the authors obtained crystal structural data and showed that a water-mediated hydrogen bond network links R288 to T0070907 but not to GW9662. NMR analysis showed that T0070907 induced two long-lived structural conformations, one similar to that induced by GW9662 and another one being unique in resembling co-repressor-bound state. Overall the studies reveal some interesting and unique SAR which is valuable for researchers working in the nuclear receptor field. Reviewers (including Reviewer 2) asked some very reasonable questions. If answered in a more appropriate ways, some confusion might be clarified and the manuscript improved to be publishable.

My assessments focus on the responses of the authors to reviewer 2's comments.

1. In the absence of exogenous agonist, does transfection of full length PPAR γ into cells induce basal activity, and if yes, to what extent?

In cell-based assays, any full length nuclear receptors, when transfected, may cause certain increase in basal activity, in the absence of exogenous agonists. The basal activity may be caused by the constitutive activity of certain receptors such as CAR, or by agonists existing in the assay systems such as serum. The level of basal activity is also likely determined by the levels of receptor being expressed in the transient transfection. (a) To distinguish constitutive activity from basal activity induced by components of the assay system, the authors should clearly describe their assay media (which induced substantial basal activity as shown in Figure S2) and compare to published data which show lower or no basal activity. Since there is no structural evidence that PPAR γ is constitutively active, the high basal activity the authors observed in Figure S2 is likely induced by components from their assay system. Indeed the authors thought that the basal activity might come from lipids and fatty acids which act as PPAR γ agonists. (b) To meaningfully compare the levels of basal activity as described in Figure S2 to published data, it will be necessary to compare the amount of plasmids used in the transient transfection and the protein levels achieved in the transient transfection.

2. Antagonist or inverse agonist?

If the high basal activity is caused by components from the assay systems (this is likely the case), then T0070907 might just be a more potent antagonist than GW9662. In a direct receptor binding

assay, does T0070907 bind with higher affinity than GW9662 to PPAR γ ? The inhibition of basal activity in cell-based assay might also be because T0070907 is more toxic than GW9662. Did the authors evaluate the cellular toxicity of both compounds in the cells used for the assays? The definition of an inverse agonist might be confusing. To some scientists (including this reviewer), an inverse agonist is for a receptor that is constitutively active (i.e., active in the absence of an agonist) – an inverse agonist will reduce the constitutive activity of the receptor. For a receptor that is not constitutively active, the basal activity is induced by agonists from the assay systems (e.g., serum, lipids or fatty acids). In this case the inhibitor (e.g., T0070907 for PPAR γ) acts as an antagonist to block the action of the agonists from the assay systems. Compared to GW9662, T0070907 more potently inhibits the basal activity of PPAR γ , by recruiting co-repressor. I suggest that instead of focusing on the difference of an “antagonist” from an “inverse agonist”, the authors may want to focus on the differential activity of the two compounds, and clearly describe what causes such difference. The main cause appears to be the unique water-mediated hydrogen bond for T0070907 which induces an additional and unique long-lived structural conformations resembling co-repressor-bound state. How about other possible causes, such as differential affinity of direct binding to the receptor, and differential cellular cytotoxicity? In addition, addressing the following question will help the readers understand the value of the studies: if PPAR γ is not a constitutively active receptor, and the basal activity is indeed induced by agonists from the assay systems (which vary depending on the exact assay conditions), what is the significance of the unique activity of T0070907 (in recruiting corepressor and reducing basal activity)?

We thank the Reviewers for their time and constructive comments regarding our manuscript. Our revised manuscript includes new data recommended by the reviewers and other revisions to the text that address the major points raised by the reviewers. We have also edited our manuscript as directed by the editor to conform to the editorial requirements at *Nature Communications*, and we split two of our previous figures (old Figures 1 and 6) into new figures (old Figure 1 = Figures 1 and 2; old Figure 6 = Figures 7 and 8) to reduce complexity. Below, our point-by-point response is formatted as a blue text indented paragraph with a left vertical line.

COMMENTS FOR AUTHORS

Reviewer #3 (Remarks to the Author):

In the revised manuscript, the authors responded to most of my concerns. However I still have concern about the interpretation of the titration data, which I mentioned before in the comment 4. The explanation of the chemical reaction pathway during the titrations (lines 435-539 and 454-457) is not quite convincing, as the authors did not consider the exchange rates and the molar fraction of each state enough. How about using a computer program such as NMRKIN (Günther, U. L. & Schaffhausen, J. *Biomol. NMR* 22, 201–209 (2002)) or LineShapeKin (Kovrigin, E. L. & Loria, J. P. *Biochemistry* 45, 2636–2647 (2006)) to analyze the titration data in order to determine the most likely chemical reaction pathway and clearly explain the titration data?

Authors' response: We thank the reviewer for the suggestion. We used the NMR simulation program LineShapeKin considering the exchange rates and molar fractions of the two slowly exchanging populations as recommended. As stated in our revised manuscript, we found that the data are best fit using a 4-state model where the two slowly exchanging (i.e., isomerization) receptor populations (R and R*) are capable of binding to the peptide (RL and R*L). This model also accounts for receptor isomerization in peptide-bound states ($R \leftrightarrow R^*$, and $RL \leftrightarrow R^*L$), and the fitting indicates that lower affinity NCoR1 binding to the mutual state (state A) results in a slow conformational change (isomerization) to the final NCoR1-bound state. This latter point is apparent in the NMR titration data because as the mutual conformation (state A) disappears upon NCoR1 binding, it shifts away rather than towards the unique conformation (state B) and NCoR1-bound state. This suggests an induced fit binding mechanism by which NCoR1 binding to the mutual conformation results in a conformational change to a more thermodynamically stable species. A similar 4-state model describes the T0070907/TRAP220 NMR data as well. For the GW9662-bound peptide interaction NMR data, we found that the TRAP220 interaction data was best described by a simpler ($R + L \leftrightarrow RL$) equation, whereas the NCoR interaction data was best described by a ($R + L \leftrightarrow RL \leftrightarrow R^*L$) equation where the NCoR peptide-bound form slowly isomerizes between two states (RL and R*L). We have updated the results and methods sections to include this new information; these analyses shown in updated Figures 7 and 8 for the T0070907- and GW9662-bound experiments, respectively.

A minor point:

(Figure 4F) "T007-bound" will be "T0070907-bound".

Authors' response: We have ensured that our manuscript does not use the abbreviation "T007" in place of "T0070907".

Reviewer #4 (Remarks to the Author):

Briefly, the authors found that two structurally similar compounds GW9662 and T0070907, function differently in regulating PPAR γ . While GW9662 blocks the binding of an agonist without recruiting co-repressor to decrease the basal activity of PPAR γ , T0070907 recruits co-repressor to reduce basal activity. To explain the differential activity of GW9662 and T0070907 on basal PPAR γ activity, the authors obtained crystal structural data and showed that a water-mediated hydrogen bond network links R288 to T0070907 but not to GW9662. NMR analysis showed that T0070907 induced two long-lived structural conformations, one similar to that induced by GW9662 and another one being unique in resembling co-repressor-bound state. Overall the studies reveal some interesting and unique SAR which is valuable for researchers working in the nuclear receptor field. Reviewers (including Reviewer 2) asked some very reasonable questions. If answered in a more appropriate ways, some confusion might be clarified and the manuscript improved to be publishable.

My assessments focus on the responses of the authors to reviewer 2's comments.

1. In the absence of exogenous agonist, does transfection of full length PPAR γ into cells induce basal activity, and if yes, to what extent? In cell-based assays, any full length nuclear receptors, when transfected, may cause certain increase in basal activity, in the absence of exogenous agonists. The basal activity may be caused by the constitutive activity of certain receptors such as CAR, or by agonists existing in the assay systems such as serum. The level of basal activity is also likely determined by the levels of receptor being expressed in the transient transfection.

(a) To distinguish constitutive activity from basal activity induced by components of the assay system, the authors should clearly describe their assay media (which induced substantial basal activity as shown in Figure S2) and compare to published data which show lower or no basal activity. Since there is no structural evidence that PPAR γ is constitutively active, the high basal activity the authors observed in Figure S2 is likely induced by components from their assay system. Indeed the authors thought that the basal activity might come from lipids and fatty acids which act as PPAR γ agonists.

Authors' response: We appreciate the reviewer's insight and as recommended we performed the luciferase reporter assay using cells cultured in media supplemented

with charcoal stripped FBS instead of “normal” FBS to determine the influence of components in the cell culture media on the basal activity of PPAR γ and the observed activities of the synthetic ligands. We found that:

- Transfection of PPAR γ into cells cultured in charcoal stripped FBS still showed increased luciferase activity relative to empty control plasmid.
- The activating and repressive ligand activity profiles were the same in normal FBS and charcoal stripped FBS.

These results (shown in updated Supplementary Figure 2) indicate that the components of the cell culture media do not have a significant influence on the pharmacological activities of the activating and repressive synthetic ligands that we tested.

(b) To meaningfully compare the levels of basal activity as described in Figure S2 to published data, it will be necessary to compare the amount of plasmids used in the transient transfection and the protein levels achieved in the transient transfection.

Authors’ response: In the luciferase reporter assay we previously reported in Figure 1B, we performed a batch transfection such that all ligand treated cells originated from the same batch of transfected cells, which were subsequently plated in 384-well plates and then treated with ligands. As suggested by the reviewer, to determine the protein levels achieved in the transient transfection, we again performed a batch transfection, subsequently plated the transfected cells in 6-well plates and treated the cells with the same ligands used in our luciferase reporter assay. Western blot analysis revealed that PPAR γ protein levels are unchanged as a result of ligand and control treatment (new Supplementary Figure 3). Thus, the different ligand transcriptional activity levels are not a consequence of different protein levels arising from protein degradation for example.

2. Antagonist or inverse agonist? If the high basal activity is caused by components from the assay systems (this is likely the case), then T0070907 might just be a more potent antagonist than GW9662. In a direct receptor binding assay, does T0070907 bind with higher affinity than GW9662 to PPAR γ ?

Authors’ response: As suggested by the reviewer, we compared the potency of GW9662 and T0070907 using several direct receptor binding assays (new Supplementary Figure 7). These ligands covalently bind to the PPAR γ LBD and would therefore be expected to show potency values near the relative protein concentration in the assays. We performed a biochemical TR-FRET ligand displacement assay that detects displacement of a fluorescent tracer ligand (Fluormone™ Pan-PPAR Green; Thermo Fisher) to determine ligand K_i (potency) values. We also performed TR-FRET coregulator interaction assays using the same TRAP220 and NCoR peptides used in our FP-based coregulator profiling assay to determine ligand EC_{50}/IC_{50} (potency) values. In these assays, GW9662 and T0070907 show similar potency values, indicating the relative ligand potencies (which as

expected are near the protein concentration used in the assays) do not influence overall functional activities of the compounds.

The inhibition of basal activity in cell-based assay might also be because T0070907 is more toxic than GW9662. Did the authors evaluate the cellular toxicity of both compounds in the cells used for the assays?

Authors' response: As suggested by the reviewer, we assessed GW9662 and T0070907 in using a CellTiter-Glo assay. All of the ligands tested at 5 μ M (the same concentration used in the other cellular assays) performed similar to DMSO control treated cells (new Supplementary Figure 4), indicating the compounds are not cytotoxic.

The definition of an inverse agonist might be confusing. To some scientists (including this reviewer), an inverse agonist is for a receptor that is constitutively active (i.e., active in the absence of an agonist) – an inverse agonist will reduce the constitutive activity of the receptor. For a receptor that is not constitutively active, the basal activity is induced by agonists from the assay systems (e.g., serum, lipids or fatty acids). In this case the inhibitor (e.g., T0070907 for PPAR γ) acts as an antagonist to block the action of the agonists from the assay systems. Compared to GW9662, T0070907 more potently inhibits the basal activity of PPAR γ , by recruiting co-repressor. I suggest that instead of focusing on the difference of an “antagonist” from an “inverse agonist”, the authors may want to focus on the differential activity of the two compounds, and clearly describe what causes such difference. The main cause appears to be the unique water-mediated hydrogen bond for T0070907 which induces an additional and unique long-lived structural conformations resembling co-repressor-bound state. How about other possible causes, such as differential affinity of direct binding to the receptor, and differential cellular cytotoxicity?

Authors' response: The reviewer raises the question of whether PPAR γ possess constitutive activity and whether a ligand that cause repression of PPAR γ over basal activity could therefore be defined as an inverse agonist. We believe the data summarized in the response above rules out the potential influences listed by the reviewer. We believe that the strongest evidence in support of defining T0070907 as an inverse agonist is the biophysical and structural data presented in our manuscript. Unlike agonist ligands, which induce a structural state that increases the affinity for coactivators and decreases the affinity for corepressors – we show that T0070907 binding induces a structural state that increases the affinity for corepressors (and lowers the affinity for coactivators) as compared to lipid-free “apo-PPAR γ . That is, the corepressor affinity is stronger (and coactivator affinity is weaker) for T0070907-bound PPAR γ vs. apo-PPAR γ . While defining the function of apo-PPAR γ is relatively straightforward using *in vitro* biochemical and structural methods, we acknowledge that the nature of the cellular experiments makes it difficult to completely rule out any other contributions from any other potential endogenous ligands present in the cells.

We do agree with the reviewer that our previous definitions for ligand activity were confusing, more specifically our use of “direct antagonist” to describe a repressive PPAR γ ligand that weakens both coactivator and corepressor affinity. In our revised manuscript, we adopted the commonly used nomenclature based on the ligand transcriptional activities in cells:

- Agonists: activate transcription over basal receptor activity.
- Inverse agonists: repress transcription over basal receptor activity.
- Antagonists: ligands that bind but do not affect transcription over basal receptor activity.

Importantly, a number of studies that have studied repressive PPAR γ ligands, which we cite in our manuscript, have also described repressive PPAR γ ligands as an inverse agonists. This includes a recent study (Goldstein et al., 2017; ref. 12) that described the activities of T0070907, SR10221, and other PPAR γ compounds in relation to targeting PPAR γ in bladder cancer. The papers cited in our manuscript that describe repressive ligands as inverse agonists include refs. 5, 6, 14, 20, 22, 43, and 44. Furthermore, the original report of T0070907 (ref. 22) also refers to T0070907 as an inverse agonist within the manuscript. There needs to be a different in nomenclature to describe ligands like GW9662 and SR1664 (i.e., antagonists) that bind but do not cause any change in PPAR γ transcription compared to basal activity; vs. ligands like T0070907 or SR10221 (i.e., inverse agonists) that repress PPAR γ transcription relative to basal activity. Based on the past use of inverse agonist in the PPAR γ field to describe repressive ligands, as well as studies of other PPARs (including β/δ ; pubmed ID 23208498), we believe it would be confusing to not call T0070907 an inverse agonist; calling it an antagonist would indicate it affects PPAR γ transcription similar to GW9662 and SR1664, which is not the case.

Finally, as suggested by the reviewer, we have focused on the structural aspects of how the R288/water/T0070907 interaction causes “corepressor-selective inverse agonism” vs. antagonism of GW9662. This should help to differentiate the corepressor-selective mechanism vs. the apparent inhibition of coregulator binding caused by the SR10221 inverse agonist, which as we describe in our manuscript is elicited through a distinct “AF-2 clash” structural mechanism.

In addition, addressing the following question will help the readers understand the value of the studies: if PPAR γ is not a constitutively active receptor, and the basal activity is indeed induced by agonists from the assay systems (which vary depending on the exact assay conditions), what is the significance of the unique activity of T0070907 (in recruiting corepressor and reducing basal activity)?

Authors’ response: As suggested by the reviewer, we have added a new section to the discussion (second to last paragraph) with information in support of constitutive PPAR γ activity relative to the corepressor-selective activity of T0070907 described by us and

reported by others. However, we also mention that if PPAR γ lacks constitutive activity, our results should only require a minor change in nomenclature, as the corepressor-selective activity of T0070907 is well supported by structural and functional evidence provided by us and by others.

REVIEWERS' COMMENTS:

Reviewer #3 (Remarks to the Author):

The description of the titration data is now adequate. There is no further concern except one minor point below.

line 464 (p. 16)

"Figure 12D" should be "Supplementary Figure 12D".

Reviewer #4 (Remarks to the Author):

The authors have addressed the points raised by this reviewer.